# Fast online node labeling with graph subsampling

## Abstract

Large data applications rely on storing data in massive, sparse graphs with millions to trillions of nodes. Graph-based methods, such as node prediction, aim for computational efficiency regardless of graph size. Techniques like localized approximate personalized page rank (APPR) solve sparse linear systems with complexity independent of graph size, but is in terms of the maximum node degree, which can be much larger in practice than the average node degree for real-world large graphs. In this paper, we consider an *online subsampled APPR method*, where messages are intentionally dropped at random. We use tools from graph sparsifiers and matrix linear algebra to give approximation bounds on the graph's spectral properties ($O(1/\epsilon^2)$ edges), and node classification performance (added $O(n\epsilon)$ overhead).

## 1 Introduction

Large data applications like search and recommendation systems, rely on data stored in the form of very large, sparse, irregular graphs, where the number of nodes can be on the order millions, billions, or even trillions. In such cases, graph-based methods, such as node prediction, must accomplish their tasks *exclusively using local operations*, e.g. where the memory complexity is independent of graph size. This is the intention of methods like the localized approximate Personalized Page Rank method (APPR) (Andersen et al., 2006; Page et al., 1999), which approximates solving a sparse linear system by truncating messages whenever the residual of that coordinate is small. However, the complexity for these methods is often in terms of the maximum node degree, and their benefits are tied to the assumption that the graph node degrees are relatively uniform.

In this work, we consider a simple solution to this problem, where high-degree nodes subsample their neighbors in message-passing methods. This solution can save considerable memory overhead when the graph's node degree distribution is heavily skewed. (See, for example, Figure 1.) However, the disadvantage of this strategy is that the stochasticity leads a high variance between each trial; thus, we implement a mechanism for grounding the residual at each iteration, to reduce this variance.

We then evaluate this method on two APPR downstream tasks, one supervised and one unsupervised. In online node labeling, future node labels are predicted using the revealed labels of the past; the APPR method is used here to solve a linear system, with a carefully tuned right-hand-side as motivated by the graph regularization method of Belkin et al. (2004), and the relaxation method of Rakhlin et al. (2012). In unsupervised clustering, we use the APPR method to acquire an improved similarity matrix, which is then clustered using nearest neighbors.

**Contributions.** In this paper, we extend the work of (Andersen et al., 2006; Zhou et al., 2023) to graphs with unfavorable node degrees, using edge subsampling. We propose a simple approach: for a threshold $\bar{q}$, we identify all nodes with degree exceeding $\bar{q}$, and subsample their neighboring edges until they have $\leq \bar{q}$ neighbors. The remaining edges are reweighted such that the expected edge weights are held consistent. In offline graphs, sparsifications of this kind require at least one full sweep through the graph, and grows with $n$, where $n$ is the number of nodes. However, online sparsification reduces this dependency on $n$, depending solely on the neighborhood structure of the query node. Our contributions are

- We give a variance reduced subsampling APPR strategy which incrementally updates the primal variable (such as in iterative refinement), producing a stable higher-accuracy estimate for very little overhead.
- We give concentration bounds on the learning performance in offline sparsification, which are comparable to that of optimal sparsifiers in previous literature.
- In the case of online sparsification, we give high probability guarantees that the method will not stop early, and show a $O(1/T)$ convergence rate overall and a linear convergence rate in expectation.
- We show superior numerical performance of online node labeling and graph clustering when subsampled APPR is integrated.

## 1.1 Related works

### 1.1.1 Applications.

We investigate two primary node labeling applications. In (supervised) online node labeling, the nodes are visited one at a time, and at time $t$, one infers the $t$th node using the revealed labels of nodes $y_1, ..., y_{t-1}$. (This models interactive applications such as online purchasing or web browsing.) In (unsupervised) graph clustering, the graph nodes are preemptively grouped, using some standard clustering method over node embeddings learned through APPR.

**Online node prediction.** This problem category has been investigated by Belkin et al. (2004); Zhu et al. (2003); Herbster & Pontil (2006); Rakhlin & Sridharan (2017), and many others. Using $z_t$ as a vector containing the already revealed label, then the solution to the APPR system is an approximation of those offered in Belkin et al. (2004); Zhu et al. (2003), for choices of regularization and interpolation. Relatedly, node prediction via approximate Laplacian inverse is also related to mean field estimation using truncated discounted random walks (Li et al., 2019), with known performance guarantees. The series of papers (Herbster & Pontil, 2006; Herbster et al., 2005) directly attack the suboptimality in online learning when applied to graphs with large diameters. They show that for this class of problems, using direct interpolation, the worst-case rate cannot be sublinear, and suggest additional graph structures to assist in this regime. Finally, Rakhlin & Sridharan (2017) tackled the problem of computing an online learning bound by offering a method, which can be seen as a modified right-hand-side of the linear system in Belkin & Niyogi (2003). The advantage of Rakhlin & Sridharan (2017) is that it provides a means of computing a learning regret bound. In Zhou et al. (2023), the analysis was tightened to $O(\sqrt{n})$ sublinear regret bounds, under the appropriate choice of kernel.

**Node embeddings and graph clustering.** The idea of learning node embeddings over large graphs has now been well studied, with tools like node2vec (Grover & Leskovec, 2016) and DeepWalk (Perozzi et al., 2014); see also Garcia Duran & Niepert (2017). Using the Laplacian as an eigenmap is also classical (Belkin & Niyogi, 2003; Weinberger et al., 2004). It is also the use of PPR vectors as node embeddings that drove the original APPR method (Andersen et al., 2006), and has been extended to more applications such as clustering (Guo et al., 2024) and improving GNNs (Bojchevski et al., 2020).

### 1.1.2 Primary tool.

In both applications, the primary tool is to quickly and efficiently solve a linear system where the primary matrix is the sparse graph Laplacian. Specifically, in many past works, the cost of solving this linear system is not accounted for in the method's complexity analysis, with the justification that there are other existing methods for an offline linear system solve; for example, the combination of offline sparsifiers (Spielman & Srivastava, 2008) and fast iterative methods (Saad, 1980). Such a method reduces the $O(n^3)$ cost of direct linear systems solve to $O(n \log(n)/\epsilon^2) + \tilde{O}(n)$, which is a significant reduction. (Here, $n$ is the number of nodes in the graph). *However, for large enough $n$, any dependency on $n$ makes the method intractable.*

**Local methods for linear systems.** While Page et al. (1999) presented the infamous Personalized PageRank (PPR) method, Andersen et al. (2006) provided the analysis that showed that, for linear systems involv-

ing graph Laplacians, adding mild thresholding and a specific choice of weighting would ensure a bounded sparsity on *all* the intermediate variables used in computation; moreover, this bound was independent of graph size (e.g. number of nodes or edges), and depended only on node degree. Fountoulakis et al. (2019) also connected this method with $\ell_1$ penalization of a quadratic minimization problem, which offered a variational perspective on the sparsifying properties of APPR. The analysis was also applied to node prediction in Zhou et al. (2023). However, in all cases, complexity bounds depend on node degree, and are only optimal when node degree is independent of graph size.

**Subsampling and sparsification.** Our main contribution is in the further acceleration of APPR through "influencer subsampling", where high degree nodes are targetted for subsampling in order to reduce the memory complexity of single-step message propagation. The analysis of this follows from prior work in graph sparsification. Specifically, existing offline graph sparsification methods include Karger (1994), which showed that random subsampling maintained cut bounds, with complexity $\tilde{O}(m + n/\epsilon^3)$; Benczúr & Karger (2015), who reduced this to $\tilde{O}(m \log^2(n))$ complexity and $O(n \log n/\epsilon^2)$ edges by subsampling less edges with estimated smaller cut values; and Spielman & Srivastava (2008); Spielman (2010); Spielman & Teng (2014) who used the principle of effective resistance to define subsampling weights to obtain optimal spectral bounds. In practice, the last two approaches are not feasible without additional randomized approaches, as computing cuts and effective resistances exactly also involve computing a large matrix pseudoinverse. More recently, Saito & Herbster (2023) investigated estimating this resistance by creating a coordinate spanning set, which is closely related to our proposed scheme, and applied it to spectral clustering.

## 2 Using graphs for node labeling

**Notation.** Denote a graph as $\mathcal{G}(\mathcal{V}, \mathcal{E}, \boldsymbol{A})$ where $\mathcal{V} = \{1, ..., n\}$ are the $n$ nodes, $\mathcal{E} \subset \mathcal{V} \times \mathcal{V}$ is the set of edges in $\mathcal{G}$, and $\boldsymbol{A}$ is the adjacency matrix for the unweighted graph (e.g. $\boldsymbol{A}_{u,v} > 0$ contain the edge weights whenever $(u, v) \in \mathcal{E}$). Denote $n = |\mathcal{V}|$ the number of nodes and $m = |\mathcal{E}|$ the number of edges. Denote $d_u = \sum_v \boldsymbol{A}_{u,v}$ the degree of node $u$ and $\boldsymbol{D} = \mathbf{diag}(d)$, and the neighbors of a node $u$ as $\mathcal{N}(u) = \{v : (u, v) \in \mathcal{E}\}$, and for a subset of nodes $\mathcal{S} \subseteq \mathcal{V}$, we express its volume as $\mathbf{vol}(\mathcal{S}) = \sum_{u \in \mathcal{S}} d_u$. For vectors, $\mathbf{supp}(\boldsymbol{x})$ is the set of indices $i$ where $x_i \neq 0$.

Now assume that each node has a ground truth binary label $y_i \in \{1, -1\}$, for $i \in \mathcal{V}$. For instance, nodes may represent customers, node label whether or not they will purchase a specific product. Consequently, the edges could reflect similarities or shared purchasing behaviors between customers. Thus, the likelihood of two connected nodes sharing the same label is high, making it possible to infer the label of an unobserved node based significantly on its neighbors.

**Simple baseline: Weighted majority algorithm.** Suppose that a small subset of the node labels are revealed, e.g. in a vector $\tilde{\boldsymbol{y}} = (\tilde{y}_1, ..., \tilde{y}_n) \in \mathbb{R}^n$ where $\tilde{y}_t \in \{-1, 1\}$ if node $t$'s label is revealed , and 0 otherwise. The revealed labels can be equal to, or approximate, the ground truth labels $\boldsymbol{y} = (y_1, ..., y_n) \in \mathbb{R}^n$, $y_t \in \{-1, 1\}$. Then, a simple method for inferring the label of an *unseen* node $t$ is to take the weighted average of its neighbors, e.g.

$$\hat{y}_t = \text{sign}\left(\frac{\sum_{i \in \mathcal{N}(t)} \boldsymbol{A}_{i,t} \tilde{y}_i}{\sum_{j \in \mathcal{N}(t)} \boldsymbol{A}_{j,t}}\right) = \text{sign}\left(\boldsymbol{e}_t^T \boldsymbol{A} \boldsymbol{D}^{-1} \tilde{\boldsymbol{y}}\right).$$

**Message passing and graph Laplacians.** We may generalize this further by expanding to the $K$-hop neighbors of $t$, using a discount factor $\beta^k$ where $k$ is the neighbor's hop distance to $t$, e.g. for $\hat{y}_t = \text{sign}(\hat{y}_{t,\text{soft}})$, where for a transition matrix $\boldsymbol{T} = \boldsymbol{A} \boldsymbol{D}^{-1}$

$$\hat{y}_{t,\text{soft}} = \sum_{k=0}^{K} \sum_{i \in \mathcal{N}(t)} (\beta^{k-1} \boldsymbol{T}^k)_{i,t} \tilde{y}_i = \beta^{-1} \sum_{k=0}^{K} \boldsymbol{e}_t^T (\beta \boldsymbol{A} \boldsymbol{D}^{-1})^k \tilde{\boldsymbol{y}} \overset{K \to \infty}{=} \beta^{-1} \boldsymbol{e}_t^T (\boldsymbol{I} - \beta \boldsymbol{A} \boldsymbol{D}^{-1})^{-1} \tilde{\boldsymbol{y}}. \tag{1}$$

(Here, we take the sum from $k = 0$ without impunity since $\tilde{\boldsymbol{y}}_t = 0$.) That is to say, a simple baseline motivates that a *globally informative, discounted estimate* of the node labels can be written as the solution

to the linear system

$$\beta^{-1}(\boldsymbol{I} - \beta\boldsymbol{A}\boldsymbol{D}^{-1})\hat{\boldsymbol{y}} = \tilde{\boldsymbol{y}}. \tag{PPR}$$

This linear system is equivalent to the well-studied linear system of the personal page-rank (PPR) vectors, which first debued in web search engines (Page et al., 1999). A symmetrized version of the PPR system

$$\underbrace{(\boldsymbol{I} - \beta\boldsymbol{D}^{-1/2}\boldsymbol{A}\boldsymbol{D}^{-1/2})}_{=:\boldsymbol{L}}\hat{\boldsymbol{y}} = \tilde{\boldsymbol{y}}. \tag{PPR-symm}$$

is more commonly studied in machine learning works (Rakhlin et al., 2012; Belkin et al., 2004). Here, the weighted Laplacian matrix $\boldsymbol{L}$ is symmetric positive semidefinite. Since the symmetrized linear system and the original PPR system are equivalent under a rescaling of the vectors $\hat{\boldsymbol{y}}$ and $\tilde{\boldsymbol{y}}$ by $\boldsymbol{D}^{1/2}$, one can view this symmetrized version as a further weighting of the seen labels, so that high-degree nodes have a tapered influence. In this work, we focus on solving this (PPR-symm) system.

## 2.1 Approximate Personalized Page Rank

The APPR method (Andersen et al., 2006), (Alg. 1) is a memory-preserving approximation of PPR. Specifically, by having proper termination steps in the PUSH substep, the method offers a better tradeoff between learnability and memory locality. It solves the linear system equivalent using operations similar to that of the Power method, to that of (PPR-symm), with $\beta = \frac{1-\alpha}{1+\alpha} \in (0,1)$ as the *teleportation parameter*.

$$\underbrace{\left(\boldsymbol{I} - \frac{1-\alpha}{1+\alpha}(\boldsymbol{D}^{-1/2}\boldsymbol{A}\boldsymbol{D}^{-1/2})\right)}_{\boldsymbol{Q}}\boldsymbol{x} = \underbrace{\frac{2\alpha}{1+\alpha}\boldsymbol{D}^{-1/2}\boldsymbol{e}_s}_{\boldsymbol{b}}, \qquad \boldsymbol{z} = \frac{1+\alpha}{2\alpha}\boldsymbol{D}^{1/2}(\boldsymbol{b} - \boldsymbol{Q}\boldsymbol{x}). \tag{2}$$

where, given the solution $\boldsymbol{x}$, the variable $\pi = \boldsymbol{D}\boldsymbol{x}$ is the desired PPR vector in web applications. The matrix $\boldsymbol{Q}$ is symmetric positive semidefinite, with eigenvalues in the range $[2\alpha/(1+\alpha), 1]$. Note that Alg. 1 assumes that the right-hand side is $\boldsymbol{e}_s$; therefore solving a full linear system with a dense right-hand side requires $n$ APPR calls, accumulated through linearity.

In Alg. 1, Steps 4 to 7 can be summarized as a PUSH operation, because of its effect in "pushing" mass from the residual to the variables. The residual vector $\boldsymbol{z} = \frac{1+\alpha}{2\alpha}\boldsymbol{D}^{1/2}(\boldsymbol{b} - \boldsymbol{Q}\boldsymbol{x}^{(t)})$ is used to identify which node messages to "push forward", and which to terminate. In Fountoulakis et al. (2019), this method is shown to be closely related to the Fenchel dual of minimizing

$$f(\boldsymbol{x}) = \frac{1}{2}\boldsymbol{x}^T\boldsymbol{Q}\boldsymbol{x} + \boldsymbol{x}^T\boldsymbol{b} + \epsilon\|\boldsymbol{x}\|_1;$$

thus the steps of APPR can be viewed as directly promoting $\ell_1$-regularized sparsity (or, in the language of large graphs, memory locality).

The following Lemma from Andersen et al. (2006) form the basis of the memory locality guarantee of this method. The steps within the while loop are often referred to collectively as the PUSH operation.

---

**Algorithm 1** APPR($\mathcal{G}$, $s$, $\epsilon$)(Andersen et al., 2006)

**Require:** $\mathcal{G} = (\mathcal{V}, \mathcal{E}, \boldsymbol{A})$, starting node $s$, tolerance $\epsilon$
1: **Init**: $\boldsymbol{x}^{(0)} \leftarrow \boldsymbol{0}$, $\boldsymbol{z}^{(0)} \leftarrow \boldsymbol{e}_s$, $t \leftarrow 1$, $\mathcal{S}^{(0)} = \{s\}$
2: **while** $\mathcal{S}^{(t)} \neq \emptyset$ **do**
3:     Pick $u \in \mathcal{S}^{(t-1)}$
4:     $\boldsymbol{x}_u^{(t)} \leftarrow \boldsymbol{x}_u^{(t-1)} + \alpha \cdot \frac{\boldsymbol{z}_u^{(t-1)}}{\sqrt{d_u}}$
5:     **for** $v \in \mathcal{N}(u)$ **do**
6:         $\boldsymbol{z}_v^{(t)} \leftarrow \boldsymbol{z}_v^{(t-1)} + \frac{(1-\alpha)\boldsymbol{A}_{u,v}}{2d_u}\boldsymbol{z}_u^{(t-1)}$
7:     $\boldsymbol{z}_u^{(t)} \leftarrow \frac{(1-\alpha)}{2}\boldsymbol{z}_u^{(t-1)}$
8:     $\mathcal{S}^{(t)} = \{v : v \in \mathcal{S}^{(t-1)} \cup \mathcal{N}(u), |\boldsymbol{z}_v| \geq d_v\epsilon\}$
9:     $t \leftarrow t + 1$
10: **Return** $\boldsymbol{x}^{(t)}$

---

**Lemma 2.1** (Monotonicity and conservation (Andersen et al., 2006)). *For all $t$, $\boldsymbol{x}^{(t)} \geq 0$, $\boldsymbol{r}^{(t)} \geq 0$. Moreover,*

$$\|\boldsymbol{x}^{(t+1)}\|_1 \geq \|\boldsymbol{x}^{(t)}\|_1, \quad \|\boldsymbol{r}^{(t+1)}\|_1 \leq \|\boldsymbol{r}^{(t)}\|_1, \|\boldsymbol{z}^{(t)}\|_1 + \|\boldsymbol{D}^{1/2}\boldsymbol{x}^{(t)}\|_1 = 1 \tag{3}$$

*for $\boldsymbol{z}^{(t)} = \frac{1+\alpha}{2\alpha}\boldsymbol{D}^{1/2}\boldsymbol{r}^{(t)}$. And, denoting $\mathbf{supp}(\boldsymbol{x})$ as the support of $\boldsymbol{x}$ (e.g. the set of indices $i$ where $x_i \neq 0$),*

$$\mathbf{supp}(\boldsymbol{x}^{(t)}) \subseteq \mathbf{supp}(\boldsymbol{x}^{(t+1)}) \subseteq \mathbf{supp}(\boldsymbol{x}^*).$$

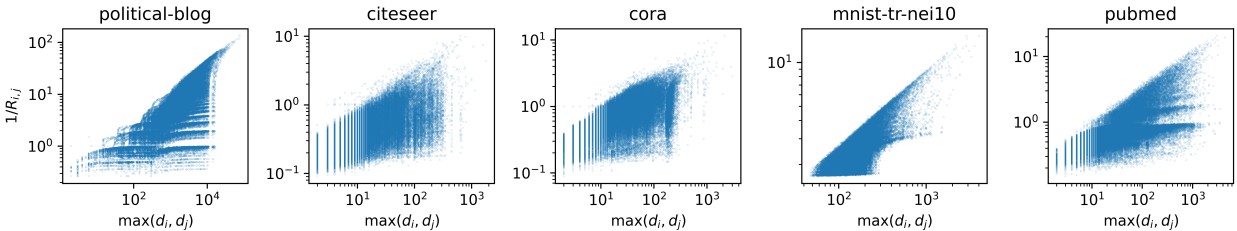

Figure 2: **Is it better to subsample influencers?** Correlation of inverse resistive distance (conductance, spectrally optimal) with high degree connecting edge is largely positive, across several graphs.

In other words, *the memory complexity of the intermediate variables $\boldsymbol{x}^{(t)}$ is bounded by that of its final solution $\boldsymbol{x}^*$.* Note that this is not the typical case for sparse methods, such as the proximal gradient method, whose intermediate variables can be dense even if the final solution is sparse. Rather, it is the subtle interplay of $\alpha$, $\epsilon$, and the PUSH operation that maintains this quality.

**Bound on $\mathbf{supp}(\boldsymbol{r}^{(t)})$.** This auxiliary variable is also an important memory-using component. Because of the nature of the PUSH method, one can infer that

$$\mathbf{supp}(\boldsymbol{r}^{(t)}) \subseteq \mathbf{supp}(\boldsymbol{x}^{(t)}) \cup \{v : v \in \mathcal{N}(u), \ u \in \mathbf{supp}(\boldsymbol{x}^{(t)})\}.$$

In other words, for a graph with unweighted edges, the complexity of the auxiliary variable $\boldsymbol{r}^{(t)}$ is bounded by the complexity of the main variable $\boldsymbol{x}^{(t)}$

$$|\mathbf{supp}(\boldsymbol{r}^{(t)})| \le |\mathbf{supp}(\boldsymbol{x}^{(t)})| + \mathbf{vol}(\mathbf{supp}(\boldsymbol{x}^{(t)})).$$

However, this bound is sensitive to the node degrees, especially for those active in the main variable.

## 3 Influencer-targeted sparsification

Degree distributions over real world graphs are often very heavy-tailed, as demonstrated in Figure 1. For message passing methods, this presents a practical challenge: if a node transmits messages to all its neighbors at each step, truncating the number of steps might not substantially alleviate the computational burden. This is due to the significant memory requirements when propagating messages from "influencer" nodes, e.g. those with exceptionally high degrees, which serve as major hubs in the network.

To combat this, we propose two schemes: *offline sparsification*, where a new, sparsified graph is produced and used in place of the original one (in similar spirit as Karger (1994); Spielman & Srivastava (2008), etc.); and *online sparsification*, where the APPR method itself subsamples neighbors at each step. *A major goal of this paper is to show this method's consistency in learning tasks, and improved computational efficiency in practice.*

First, we ask a question: is it better to remove the edge of an "influencer" (high-degree node), or will such an action disproportionately degrade performance? To offer some preliminary intuition, we compare different sparsifiers in terms of correlation with resistive distance, a measure proposed by Spielman & Srivastava (2008) for spectrally optimal graph sparsification; and by analyzing the edge ratios of sparsified graphs to assess their alignment with node learnability.

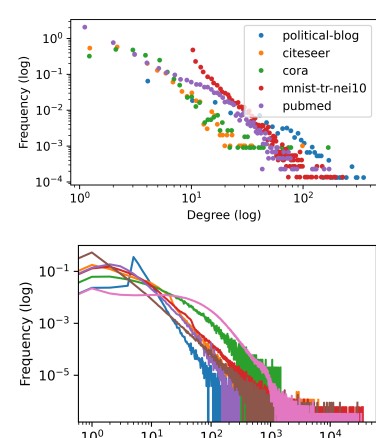

Figure 1: Degree distributions of small (top) vs large (bottom) graphs

### 3.1 Resistive distance

For an unnormalized graph Laplacian $\boldsymbol{L}_{\mathrm{un}} = \boldsymbol{D} - \boldsymbol{A}$, the resistive distance between nodes $i$ and $j$ is

$$\boldsymbol{R}_{i,j} = \boldsymbol{L}_{i,i}^{\dagger} + \boldsymbol{L}_{j,j}^{\dagger} - 2\boldsymbol{L}_{i,j}^{\dagger}.$$

In electrical engineering, this measures the effective resistance between two junctions $i$ and $j$ in a network of resistors, whose weights form the weights of the graph ($\boldsymbol{A}_{i,j} = 1/r_{i,j}$). Importantly, resistive distance is known to be the optimal metric for offline sparsification for preserving spectral properties (Spielman & Srivastava, 2008). However, in general, it is challenging to compute, as it requires a full Laplacian pseudoinverse. Figure 2 compares a graph edge's "influencer status" (e.g. highest degree connected node) by comparing the correlation between an influencer-connecting edge, and the edge's inverse resistive distance (conductance). There is a positive correlation, hinting that indeed this is a good (and cheap) metric for subsampling.

**Non-spectral properties.** Figure 3 investigates the performance of edge subsampling as it pertains to node labeling, our desired downstream task. Specifically, it gives the proportion of edges that connect different-labeled nodes over same-labeled nodes. (Smaller is better.) This suggests that influencer-based sparsity not only preserves but can sometimes improve the edge ratio despite a reduction in the number of edges (even moreso than resistive subsampling).

### 3.2 Graph sparsification

We next consider a general (offline) graph sparsification scheme, where $\mathcal{G} = (\mathcal{V}, \mathcal{E}, \boldsymbol{A})$ is sparsified into $\mathcal{G}' = (\mathcal{V}, \mathcal{E}', \boldsymbol{A}')$ where $\mathcal{E}' \subset \mathcal{E}$. This is given in Alg. 2. We consider three main forms of sparsification: uniformly removing edges (Karger, 1994), removing edges based on resistive distance (Spielman & Srivastava, 2008), and removing edges based on node degree (influencer). We first give high-concentration bounds influencer-based sparsifications, to show asymptotic consistency.

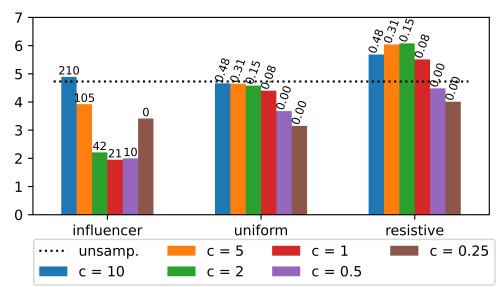

---

**Algorithm 2** General offline sparsifier

---

**Require:** Graph $\mathcal{G} = (\mathcal{V}, \mathcal{E}, \mathbf{A})$,
1: edge probabilities $p_{u,v}$ for $(u, v) \in \boldsymbol{E}$
2: Initialize $\tilde{\mathcal{E}} = \emptyset$, $\tilde{\boldsymbol{A}} = \boldsymbol{0}$, $\tilde{\mathcal{G}} = (\mathcal{V}, \tilde{\mathcal{E}}, \tilde{\boldsymbol{A}})$
3: **for** $i = 1, ..., m$ **do**
4:      Randomly pick edge $(u, v)$.
5:      With probability $p_{u,v}$,
6:      $\tilde{\boldsymbol{A}}_{u,v} = \tilde{\boldsymbol{A}}_{v,u} = \frac{1}{p_{u,v}}$, $\tilde{\mathcal{E}} = \tilde{\mathcal{E}} \bigcup \{(u, v)\}$
         **return** $\tilde{\mathcal{G}} = (\mathcal{V}, \tilde{\mathcal{E}}, \tilde{\mathbf{A}})$

---

Figure 3: **Edge ratio for political-blog.** Here, $\bar{q} = c\times$ median distribution for (inf.), and the resulting sparsity level is used for $p$ in (res.) and (unif.).

**Theorem 3.1** (Offline sparsification). *Consider $\boldsymbol{L} = \boldsymbol{I} - \boldsymbol{D}^{-1/2}\boldsymbol{A}\boldsymbol{D}^{-1/2}$, $\tilde{\boldsymbol{L}}$ the Laplacian matrices corresponding to a graph and its sparsified version, and $\bar{q}$ the subsampling threshold. Then, for any $\boldsymbol{x} \in \mathbb{R}^n$,*

$$\mathbf{Pr}(|\boldsymbol{x}^T \tilde{\boldsymbol{L}}_I \boldsymbol{x} - \boldsymbol{x}^T \boldsymbol{L}_I \boldsymbol{x}| \geq \epsilon) \leq 2 \min\{e^{-\frac{\epsilon^2}{8\|\boldsymbol{x}\|_{\infty}^2 \|\boldsymbol{x}\|_2^2}}, e^{-\frac{\epsilon^2}{8\|\boldsymbol{x}\|_{\infty}^4 |\mathcal{S}_I|}}\}.$$

For a normalized $\|\boldsymbol{x}\|_2 = O(1)$, useful values of $\epsilon \in (0, 2)$. The bound's usefulness also depends on the choice of $\boldsymbol{x}$, and the ratio $\frac{\|x\|_2}{\|x\|_{\infty}}$. For a Gaussian random vector in $\mathbb{R}^n$, this ratio is about $O(\sqrt{n/\log(n)})$.

Plugging in this trend, the right-hand side reduces to $O(\min\{e^{-\frac{\epsilon^2 n}{8\log(n)}}, e^{-\frac{\epsilon^2 n^2}{8\log(n)^2 |\mathcal{S}_I|}}\})$. Note that both terms are nontrivial for very large $n$, with stronger rates if $|\mathcal{S}_I| < O(n/\log(n))$.

**Corollary 3.1.1.** *For $n$ large enough,*

$$\mathbf{Pr}(\boldsymbol{L}_I - \epsilon \boldsymbol{I} \preceq \tilde{\boldsymbol{L}}_I \preceq \boldsymbol{L}_I + \epsilon \boldsymbol{I}) \geq 1 - 12 \exp(-\frac{\epsilon^2 n^3}{32 \log(n)}) - 4 \exp(-c' n \epsilon/2)$$

# 4 APPR with online sparsification

---

**Algorithm 3** DUALCORRECT($\mathcal{G}$, $\boldsymbol{x}$, $\bar{q}$)

---

1: Identify

$$\mathcal{U} \subseteq \mathbf{supp}(\boldsymbol{x}), \qquad |\mathcal{U}| \leq \bar{q}.$$

2: $\tilde{\boldsymbol{z}} = \boldsymbol{0}$

3: **for** $u \subset \mathcal{U}$ **do**

4:     Sample neighbors $\mathcal{S} \subset \mathcal{N}(u)$

5:     Update

$$\Delta \boldsymbol{z} = (x_u \cdot \boldsymbol{A}\boldsymbol{D}^{-1}\boldsymbol{e}_k)_{(\mathcal{S})}$$

6:     Push operation

$$\tilde{z} \quad \leftarrow \quad \tilde{z} + \frac{(1-\alpha)\sqrt{d_u}\Delta\boldsymbol{z}}{2\alpha}$$

$$\tilde{z}_u \quad \leftarrow \quad \tilde{z}_u - \frac{(1+\alpha)\sqrt{d_k}}{2\alpha}$$

7: **Return** $\tilde{\boldsymbol{z}}$

---

**Algorithm 4** PUSHAPPR($\mathcal{G}$, $\boldsymbol{x}$, $\boldsymbol{z}$, $\bar{q}$, $\mathcal{A}$)

---

1: Set $\boldsymbol{x}^{(0)} = \boldsymbol{x}$, $\boldsymbol{z}^{(0)} = \boldsymbol{z}$

2: **for** $u_i \in \mathcal{A} \neq \emptyset$ **do**

3:     Update $\boldsymbol{x}^{(i+1)} \leftarrow \boldsymbol{x}^{(i)} + \frac{\alpha}{\sqrt{d_{u_i}}}z_{u_i}^{(i)}\boldsymbol{e}_{u_i}$

4:     Sample neighbors $\mathcal{S} \subset \mathcal{N}(u_i)$, $|\mathcal{S}| \leq \bar{q}$

5:     Update

$$\Delta \boldsymbol{z}^{(i)} = (\boldsymbol{z}_{u_i}^{(i)} \cdot \boldsymbol{A}\boldsymbol{D}^{-1}\boldsymbol{e}_{u_i})_{(\mathcal{S})}$$

6:     Push operation

$$\boldsymbol{z}^{(i+1)} \quad \leftarrow \quad \boldsymbol{z}^{(i)} + \tfrac{1-\alpha}{2}\Delta\boldsymbol{z}^{(i)},$$

$$\boldsymbol{z}_{u_i}^{(i+1)} \quad \leftarrow \quad \tfrac{1-\alpha}{2}\boldsymbol{z}_{u_i}^{(i)},$$

7: **Return** $\boldsymbol{x}^{(|\mathcal{S}^{(t)}|)}$, $\boldsymbol{z}^{(|\mathcal{S}^{(t)}|)}$

---

We now consider the problem of improving Alg. 1 via *online sparsification*; e.g., the APPR method itself is modified to perform subsampling at each step. Here, the notation is defined for unbiased sampling

$$(\boldsymbol{w}_{(\mathcal{S})})_i = \frac{\boldsymbol{w}_i n}{|\mathcal{S}|} \text{ if } i \in \mathcal{S}, \qquad \tilde{\boldsymbol{w}}_i = 0 \text{ otherwise.}$$

The RANDOM-APPR algorithm (Alg 5) aims to compute an approximation of the vector $\mathbf{x}$ for a given graph $\mathcal{G}$, accuracy $\epsilon$, and start node $s$. Specifically, for a query node $u$, we subsample $\bar{q}$ of its neighbors $\mathcal{N}(u)$ whenever $|\mathcal{N}(u)| > \bar{q}$, for some threshold $\bar{q}$ we set. This is to eliminate any extra memory requirements in forming an offline sparsified graph. Additionally, an online implementation allows for more exploration. If the starting node $e_s$ is only connected to influencers, for example, an offline sparsification will likely disconnect it from the graph altogether; this is not necessarily true in online subsampling.

---

**Algorithm 5** RANDOMAPPR($\mathcal{G}$, $\boldsymbol{s}$, $\epsilon$, $\bar{q}$) with variance reduced debiasing

---

**Require:** $c < 1$, tol. $\epsilon > 0$, parameter $\alpha$

1: **Init** $\bar{\boldsymbol{x}} = \boldsymbol{x}^{(0)} = 0$, $\bar{\boldsymbol{z}} = \boldsymbol{z}^{(0)} = \boldsymbol{e}_s$

2: **for** $t = 1, \ldots$ **do**

3:     **if** Dual correct **then**

       % Estimate unbiased $\boldsymbol{z}^{(t)} = \boldsymbol{D}^{-1}\boldsymbol{Q}\boldsymbol{x}^{(t)}$

4:         $\tilde{\boldsymbol{z}} \leftarrow$ DUALCORRECT($\mathcal{G}$, $\bar{q}$, $\boldsymbol{x}^{(t)}$)

5:         $\bar{\boldsymbol{x}} \leftarrow \boldsymbol{x}^{(t)}$, $\boldsymbol{x}^{(t)} \leftarrow \boldsymbol{0}$

6:         $\bar{\boldsymbol{z}} \leftarrow \bar{\boldsymbol{z}} - \tilde{\boldsymbol{z}}$, $\boldsymbol{z}^{(t)} \leftarrow \bar{\boldsymbol{z}}$

7:     Find set $\mathcal{A}^{(t)} = \{k : |z_k^{(t)}| > cd_k\epsilon\}$

8:     **if** $\mathcal{A}^{(t)} = \emptyset$ **then**

9:         **break**

10:     $\boldsymbol{x}^{(t+1)}, \boldsymbol{z}^{(t+1)}$ =
PUSHAPPR($\mathcal{G}$, $\boldsymbol{x}^{(t)}$, $\boldsymbol{z}^{(t)}$, $\bar{q}$, $\mathcal{A}^{(t)}$).

11: **Return** $\bar{\boldsymbol{x}} + \boldsymbol{x}^{(T)}$

---

**Resampling the dual variable.** Consider Alg. 5 in which the dual correction step is completely skipped. Then, note that the PUSH steps in Alg. 5 are the same as those in Alg. 1. While in expectation, the solution $\boldsymbol{x}^{(T)}$ will be that of the original, unsampled solution, in each run the dual variable $\boldsymbol{z}^{(t)}$ will drift, causing high variance in $\boldsymbol{x}$. Therefore, the primary purpose of DUALCORRECT is to provide an unbiased estimate of the dual variable $\boldsymbol{z}^{(t)}$ at each step, offering an important dual correction. This is critical to achieve similar bounds despite subsampling.

However, simply resampling the dual variable at each step is computationally expensive, since using a small sampling rate for this specific step can result in mismatch between the primal iterate $\boldsymbol{x}^{(t)}$ and the dual $\boldsymbol{z}^{(t)}$, which causes algorithmic instability. Therefore, we borrow ideas of "iterative refinement" from large-scale convex optimization solvers, and increment the primal and dual variables at each dual correction step. In other words, at each such step, the linear system is shifted from $\boldsymbol{Q}\bar{\boldsymbol{x}} = \boldsymbol{b}$ to $\boldsymbol{Q}\boldsymbol{x}^{(t)} = \boldsymbol{r}$ where $\boldsymbol{r}$ is the current computed residual. Then by accumulating $\bar{\boldsymbol{x}} = \boldsymbol{x}^{(t_1)} + \boldsymbol{x}^{(t_2)} + \cdots$ at each correction step, we return $\bar{\boldsymbol{x}}$ the

sum of the incrementally computed primal variables. In practice, this method is much more stable, and achieves very little extra complexity overhead.

## 4.1 Analysis

All proofs are given in Appendix A, B, and C.

**Theorem 4.1.** *In online APPR, for all $t$, $i \in \mathcal{S}_t$, $\mathbb{E}[\bar{z}^{(t,i)}] \geq 0$, $\mathbb{E}[x^{(t,i)}] \geq 0$ and*

$$\mathbb{E}[\tilde{z}^{(t,i+1)}] \leq \mathbb{E}[\bar{z}^{(t,i)}], \; \mathbb{E}[\bar{z}^{(t+1)}] \leq \mathbb{E}[\bar{z}^{(t)}], \quad \mathbb{E}[x^{(t,i+1)}] \geq \mathbb{E}[x^{(t,i)}], \; \mathbb{E}[x^{(t+1)}] \geq \mathbb{E}[\bar{x}^{(t)}]. \quad \text{(monotonicity)}$$

*Moreover,*

$$\|\mathbb{E}[D^{1/2}x^{(t)}]\|_1 + \|\mathbb{E}[\tilde{z}^{(t)}]\|_1 = \|D^{-1/2}b\|_1 \quad \text{(conservation)}$$

*and*

$$\|\mathbb{E}[D^{1/2}x^{(t+1)}]\|_1 - \|\mathbb{E}[D^{1/2}x^{(t)}]\|_1 = \|\mathbb{E}[\tilde{z}^{(t)}]\|_1 + \|\mathbb{E}[\tilde{z}^{(t+1)}]\|_1 \leq |\mathcal{S}_t|\alpha\epsilon. \quad \text{(descent)}$$

Next, we give convergence results.

**Assumption 4.1.** *There exists constants $\sigma^{(t)}$ and $\sigma^{(t,i)}$ for $t = 1,...,T$ and $i = 1,...,|\mathcal{S}^{(t)}|$ such that*

- *the random variable $\tilde{z}_j^{(t)}|x^{(t)}$ is subgaussian with parameter $(\sigma^{(t)})^2$, for all $j$*

- *the random variable $\tilde{z}_j^{(t,i+1)}|\tilde{z}^{(t,i)}$ is subgaussian with parameter $(\sigma^{(t,i)})^2$, for all $j$*

**Assumption 4.2.** *There exists a constant $R$ upper bounding each residual term*

$$\max\{\|\tilde{z}^{(t)}\|_\infty, \|\tilde{z}^{(t,i)}\|_\infty\} \leq R,$$

*for all $t = 1,...,T$, $i = 1,...,|\mathcal{S}_t|$.*

Note that from monotonicity, each PUSH action cannot increase $\|\tilde{z}^{(t,i)}\|_\infty$. However, the unbiased estimation of $\|\tilde{z}^{(t)}\|_\infty$ is not theoretically bounded; in practice, we observe $\|r^{(t)}\|_\infty$ to be small (usually $< 1$), so this assumption is quite reasonable. Under these assumptions, we have the following conclusions.

**Theorem 4.2** (Chance of stopping too early.). *Consider the online version of the algorithm. The probability that for some $i$, $D^{-1}z_i^{(t)} > \epsilon$ but $D^{-1}\tilde{z}_i^{(t)} < c\epsilon$ is bounded by*

$$\mathbf{Pr}(|D^{-1}z_i^{(t)}| > \epsilon, \; |D^{-1}\tilde{z}_i^{(t)}| < c\epsilon) \leq \exp\left(-\frac{(1-c)^2\epsilon^2}{2\sigma_t^2}\right).$$

That is to say, the chance of stopping too early is diminishingly small; even more so if we reduce the user parameter $0 < c < 1$.

**Theorem 4.3** (1/T rate). *Consider $f(x) = \frac{1}{2}x^T Q x - b^T x$ where $b = \frac{2\alpha}{1+\alpha}D^{-1/2}e_s$. Initialize $x^{(0)} = 0$. Define $M^{(t)} = \sum_{\tau=1}^{t}|\mathcal{S}_\tau|$ the number of push calls at epoch $t$. Then,*

$$\min_{t,j}\|\nabla f(x^{(t,j)})\|_2^2 \leq \frac{1}{\alpha M^{(t)}} + \frac{\alpha\sigma_{\max}^2}{2}$$

*where $\sigma_{\max} = \max_{i,t}\{\sigma^{(i,t)}, \sigma^{(t)}\}$. Here, $\alpha$ can be chosen to mitigate the tradeoff between convergence rate and final noise level.*

**Theorem 4.4** (Linear rate in expectation). *Using $\bar{z}^{(t)} = \frac{1+\alpha}{2\alpha}D^{1/2}(b - Qx^{(t)})$, in expectation,*

$$\|\mathbb{E}[\tilde{z}^{(t+1)}]\|_1 \leq \exp\left(-\frac{M_t\alpha\epsilon}{R}\right).$$

*Moreover, since $\|z^{(t)}\|_1 \leq \sqrt{n}\|z^{(t)}\|_2$,*

$$\mathbf{Pr}\left(\|z^{(t)}\|_1 \geq \exp\left(-\frac{M_t\alpha\epsilon}{R}\right)^{M_t} + \epsilon\right) \leq \exp\left(-\frac{\epsilon^2}{2\alpha^2\sqrt{n}\left(\sum_{\tau=1}^{t}\sum_{i\in\mathcal{S}_t}\sigma^{(t,i)} + \sum_{\tau=1}^{t}\sigma^{(t)}\right)}\right).$$

These two Theorems (4.4 and 4.3) offer two points of view of the convergence behavior: linear in expectation, and at least $O(1/T)$ deterministically.

**Algorithm 6** RELAXATION($\boldsymbol{L}, \gamma$)(Rakhlin & Sridharan (2017))

1: $n =$ size of $\boldsymbol{L}$ (num of rows or columns)
2: Compute $\boldsymbol{M} = \left(\frac{\boldsymbol{L}}{2\gamma} + \frac{\boldsymbol{I}_n}{2n}\right)^{-1}$
3: $\tau_1 = \mathbf{tr}(\boldsymbol{M}), a_1 = 0, \boldsymbol{G} = \boldsymbol{0} \in \mathbb{R}^{K \times n}$
4: **for** $t = 1, \dots, n$ **do**
5:   $z_t = -2\boldsymbol{G}\boldsymbol{M}_{:,t}/\sqrt{a_t + D^2 \cdot \tau_t}$
6:   Predict $\hat{y}_t \sim \omega_t(\boldsymbol{z}_t)$, $\nabla_t = \nabla\phi_{\boldsymbol{z}_t}(\cdot, y_t)$
7:   Update $\boldsymbol{G}_{:,t} = \nabla_t$
8:   $a_{t+1} = a_t + 2\nabla_t^\top \boldsymbol{G}\boldsymbol{M}_{:,t} + \boldsymbol{M}_{tt} \cdot \|\nabla_t\|_2^2$
9:   $\tau_{t+1} = \tau_t - \boldsymbol{M}_{tt}$
   **return** $\boldsymbol{z}_t$, $t = 1, ..., n$

**Algorithm 7** REGULARIZE($\mathbf{L}, \gamma$)(Belkin et al. (2004))

1: $n =$ size of $\boldsymbol{L}$ (num of rows or columns)
2: Compute $\boldsymbol{M} = \left(\frac{\mathbf{L}}{2\gamma} + \frac{\boldsymbol{I}_n}{2n}\right)^{-1}$
3: **for** $t = 1, \dots, n$ **do**
4:   $z_t = -2\boldsymbol{G}\boldsymbol{M}_{:,t}$
5:   Predict $\hat{y}_t \sim \omega_t(\boldsymbol{z}_t)$
6:   Update $\boldsymbol{G}_{:,t} = y_t$
   **return** $\boldsymbol{z}_t$, $t = 1, ..., n$

# 5 Applications

## 5.1 Online node labeling (ONL)

In this framework (Belkin et al., 2004; Herbster et al., 2005; Zhou et al., 2023), node labels are revealed after visiting. Formally, we traverse through the nodes in some order $t = 1, ..., n$, we infer the label $\hat{y}_t \in \{1, -1\}$, and incur some loss $\ell(y_t, \hat{y}_t)$. Then the true label $y_t$ is revealed (e.g. the customer bought the item or left the page), and we predict $\hat{y}_{t+1}$ using the now seen labels $y_1, ..., y_t$.

We integrate RANDOMAPPR with two well-studied methods for ONL:

- REGULARIZE (Alg. 6), where (PPR-symm) is solved using $\mathbf{y}_t = [y_1, ..., y_{t-1}, 0, ..., 0]^T \in \mathbb{R}^{n \times K}$.

- RELAXATION(Alg. 7), which follows the process outlined in Rakhlin & Sridharan (2017).

In both cases, the methods also depend on the graph smoothness parameter $\gamma_{\text{sm}} = \boldsymbol{y}^T \boldsymbol{L} \boldsymbol{y}$. [1] Note that $\gamma$ is then integrated into the method, such that it matches $\beta = \frac{n}{\gamma}$ as in our discount factor in Section 2.

We quantify the success of a set of predictions $\hat{\boldsymbol{y}} = [\hat{y}_1, ..., \hat{y}_n]^T \in \mathbb{R}^{n \times K}$ in terms of the solution's suboptimality, compared against the best solution that is $\gamma$-smooth, e.g.

$$\text{regret}(t) = \ell_t(\boldsymbol{y}, \hat{\boldsymbol{y}}) - \inf_{\boldsymbol{y}' \in \mathcal{F}_\gamma} \ell_t(\boldsymbol{y}, \boldsymbol{y}'), \quad \text{where} \quad \ell_t(\boldsymbol{y}, \hat{\boldsymbol{y}}) = \sum_{i=1}^t \mathbf{1}_{y_i \neq \hat{y}_i}, \ \mathcal{F}_\gamma = \{\boldsymbol{y} : \mathbf{tr}(\boldsymbol{y}^T \boldsymbol{L} \boldsymbol{y}) \leq \gamma_{\text{sm}}\}.$$

In both methods, $\omega_t(z_t)$ transforms a positive vector into a probability vector through *waterfilling*, e.g. $\hat{y}_t = \max\{0, z_t - \tau\}$ such that $\hat{y}_t$ sums to 1. The function $\phi$ is a specially designed convex loss function, introduced in Rakhlin & Sridharan (2017). Overall, the RELAXATION method, originally introduced in Rakhlin & Sridharan (2017), gives a regret bound, which is then tightened and shown to be sublinear in Zhou et al. (2023):

**Theorem 5.1** (Rakhlin & Sridharan (2017); Zhou et al. (2023)). *For some (graph-independent) constant* $D = O(K)$, *and* $\rho = \frac{\log(\gamma)}{\log(n)}$, *Alg. 6 has the regret bound where* $\gamma_{\text{sm}} = \boldsymbol{y}^T \boldsymbol{L} \boldsymbol{y}$:

$$\text{regret}(n) \overset{\overset{\textit{Rakhlin \& Sridharan (2017)}}{}}{\leq} \sqrt{\mathbf{tr}\left(\left(\frac{\boldsymbol{L}}{2\gamma_{\text{sm}}} + \frac{\boldsymbol{I}}{2n}\right)^{-1}\right)} \overset{\overset{\textit{Zhou et al. (2023)}}{}}{\leq} D\sqrt{2n^{1+\rho}}.$$

---

[1] In practice, $\gamma_{\text{sm}}$ is not a parameter known ahead of time. However, an incorrect guess of $\gamma_{\text{sm}}$ usually does not affect the practical performance significantly, and a correct guess allows us to align the numerical performance more closely with the available regret bounds.

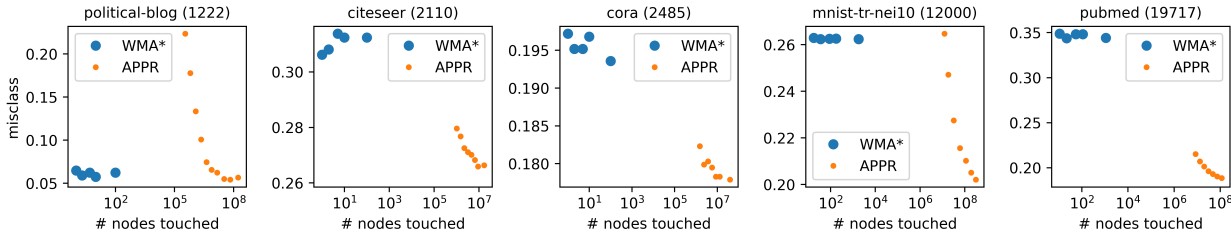

Figure 4: **Baseline comparisons.** Standard APPR versus extended WMA on ONL (WMA*). The x-axis represents memory complexity (total number of nodes queried) and the y-axis shows performance in terms of misclassification rate. While WMA operates faster, APPR is superior at higher performance levels.

*Furthermore, integrating APPR into RELAXATION adds overhead to result in*

$$\text{regret}(n) \le D\sqrt{(1+K^2)n^{1+\rho}}.$$

*where K is the number of classes.*

Specifically, in Rakhlin & Sridharan (2017), the assumption is that the matrix inversion steps are done explicitly, using full matrix linear system solvers. To tighten the analysis under practical methods, Zhou et al. (2023) computes a new regret bound where the matrix inversion steps are replaced with vanilla APPR. This theorem suggests that small perturbations in the numerics of the learning method will not greatly deteriorate the overall regret bound.

Now we consider the vanilla APPR method learned over a graph perturbation resulting in weighted Laplacian $\tilde{\boldsymbol{L}}$ has label smoothness $\tilde{\gamma}_{\text{sm}} = \mathbf{tr}(\boldsymbol{y}^T \tilde{\boldsymbol{L}} \boldsymbol{y})$.

**Theorem 5.2.** *RELAXATION over $\tilde{\boldsymbol{L}}$ using $\sigma'_{\text{sm}} = \mathbf{tr}(\boldsymbol{y}^T \tilde{\boldsymbol{L}} \boldsymbol{y})$ achieves*

$$\text{regret}(n) \le D\sqrt{2n^{1+\rho}} + \sqrt{\frac{\epsilon n}{1-\beta}}.$$

The proof is in Appendix D. The additional error maintains a sublinear regret.

### 5.2 Unsupervised clustering

Clustering is crucial for community detection, social network analysis, and other applications where the inherent structure of the data must be discovered. In this context, we use the node embeddings as the rows to the matrix inverse $\boldsymbol{Z} = (\boldsymbol{L} + \beta \boldsymbol{I})^{-1}$ via RANDOMAPPR. Then, clustering is done by first identifying the highest degree nodes as seeds $\mathcal{C} \subset \mathcal{V}$, and then assigning the cluster based on largest value in $\boldsymbol{Z}$:

$$\mathbf{cluster}(u) = \text{PICKONE}(\arg\max_{j \in \mathcal{C}}\{\boldsymbol{Z}_{t,j}\})$$

The evaluation score is the normalized sum purity of the ground truth labels, over each cluster

$$\mathbf{score} = \frac{\sum_{j \in \mathcal{C}} \mathbf{purity}(\mathcal{S}_j)}{n}, \qquad \mathcal{S}_j = \{u : \mathbf{cluster}(u) = j\}.$$

This method is reminiscent of latent semantic analysis in document retrieval; for example, by using a co-occurance matrix as a proxy for similarity (Deerwester et al., 1990).

## 6 Numerical experiments

We evaluate the various methods across several tasks. The total number of nodes visited is the main complexity measure. In each case, we explore the tradeoff between task performance, and complexity.

**Baseline.** Figure 4 compares the performance of APPR (without any subsampling) to that of simple message passing (generalized WMA, denoted WMA*, where we use (1) for finite $K \ge 1$). Generally, while WMA is more computationally efficient, APPR consistently delivers superior performance across large graphs.

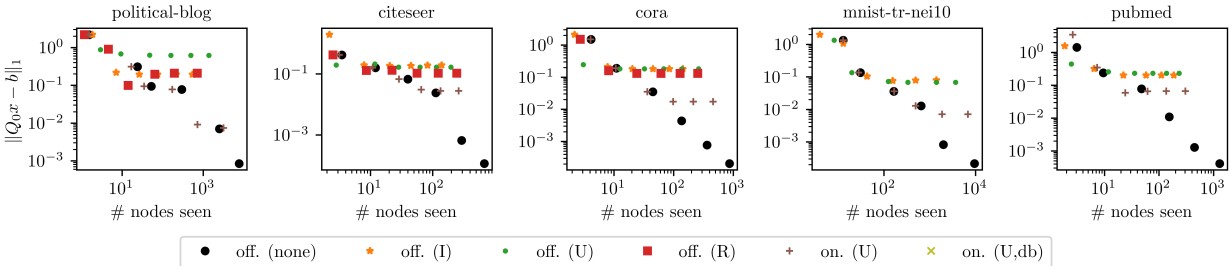

Figure 5: **APPR.** The residual of the original linear system is $\|\boldsymbol{Q}_0 x - \boldsymbol{b}\|_2$. off = offline sparsification, on = online sparsification. O = original (unsparsified). For offline, U = uniform, R = resistive, I = influencer. All online sparsifications are influencer, and further subsampled via U = uniform, W = edge weighted, D = degree weighted. c = with dual correction.

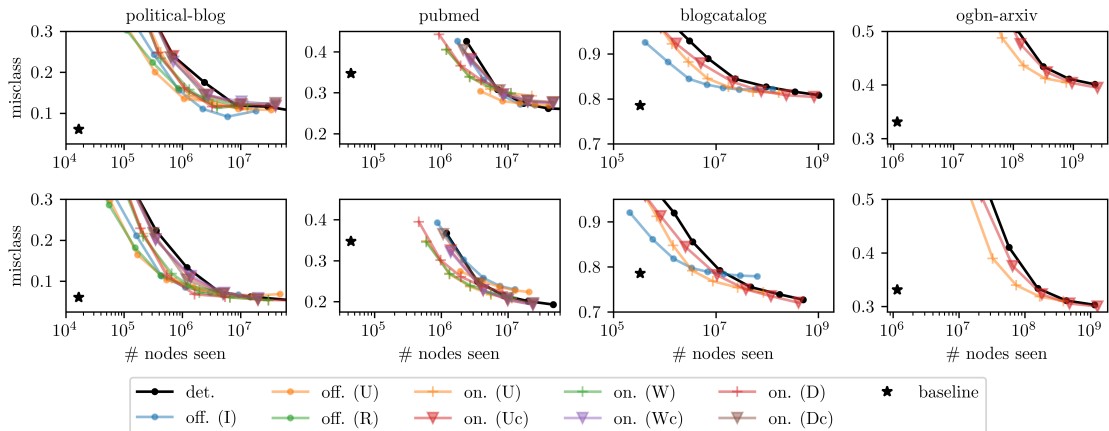

Figure 6: **ONL performance.** Top: relaxation, bottom: regularization. det = deterministic APPR, off = offline, on = online. U = uniform, W = weighted by edge, D = weighted by degree, R = weighted by resistive distance. Baseline = WMA (1-hop). All experiments were run for the same set of $\epsilon$ values. Offline graph sparsification was too memory-intensive for large graphs. For online, we only subsample influencer nodes. c = with dual correction. Lower is better.

**Offline sparsification.** Figure 5 demonstrates the performance of APPR applied to an offline sparsified graph. The first row gives the residual over the original linear system (smaller is better.) The second gives the residual over the new, biased linear system. The bias in each run is evident; the residual of the original system obtains a noise floor when subsampling is used; this is lowered first by online sparsification, and then by dual correction. Note that when measuring performance over linear systems, there is not much benefit to subsampling; the main advantage comes in learning tasks.

**ONL performance.** Figure 6 gives the performance of offline and online (with and without dual correction) methods for online node labeling. Note that for small graphs, the baseline (WMA) is very strong; however, for larger graphs, the tradeoff for reduced misclassification rate becomes more apparent. It is also clear that the RELAXATION method, although providing strong rates, is not as strong in practice as the REGULARIZATION method. Nonetheless, both methods are closely related, and their respective advantages and disadvantages stem from subtle adjustments in procedures and hyperparameters.

**Unsupervised clustering.** Finally, in Figure 7 we evaluate the performance of RANDOMAPPR in producing node embeddings, which are then used in clustering, and evaluated based on their ground truth labels. Only influencer-based uniform sampling is used. An ablative study is in Appendix E.

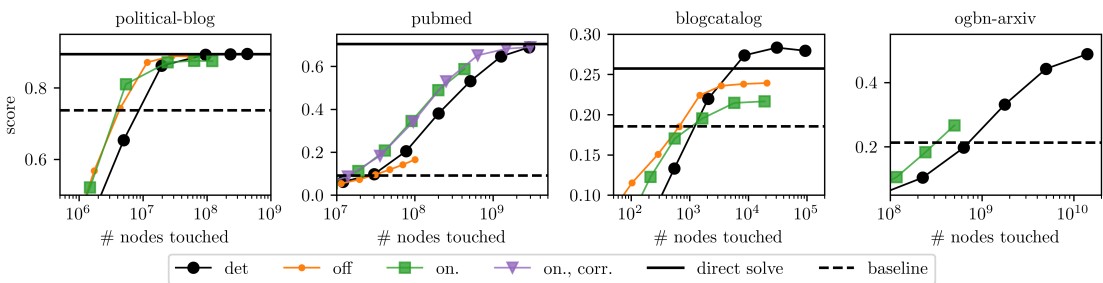

Figure 7: **Clustering performance.** det = deterministic APPR, off = offline, on = online. U = uniform. Baseline = WMA (1-hop). All experiments were run for the same set of $\epsilon$ values. Offline graph sparsification and direct solve were too memory-intensive for large graphs. For online, we only subsample influencer nodes. c = with dual correction. direct solve is shown as an upper bound, when it is computable. Higher is better.

## 7 Discussion

Our results demonstrate a tradeoff between performance and memory utilization when random sparsification is used. For very large graphs, some form of memory alleviation is mandatory, and thus such methods must be considered even if performance degrades. While offline sparsification produces significant noise in solving linear systems, it is still robust in the downstream online prediction task. Moreover, online sparsification produces good results in both linear solving and learning.

**The impact of downweighting influencers.** Figures 3 and 10 (appendix) show how different offline subsampling methods affect the edge ratio, which correlates graph clustering features with true labels. Note that influencers and inverse resistive distances are correlated, but not identical; for example, edges between nodes in a fully connected subgraph will have low resistive distance, but are not necessarily influencers. In the `political-blog` dataset (Fig 3), it seems beneficial to subsample edges from extreme influencers who may ignore political affiliations. However, because low resistive distances might also indicate cliques tied to political parties, indiscriminately removing these can be harmful. These dynamics differ across datasets and applications.

**Relationship to other locality sampling methods.** The idea of subsampling edges to reduce memory complexity in graph applications is not new; in particular, in Shin et al. (2024); Huang & Zitnik (2020); Wu et al. (2023), it is used to alleviate message passing for GNNs. However, in many of those methods, the sampling strategies focus on a hop-radius, and alleviate small-world effects via aggressive subsampling, weighted by edge weights or outgoing degrees. We argue that using an APPR-inspired approach offers a more powerful tradeoff between performance and sparsity, as it inherently does not restrict the hop-radius in fusing neighboring nodes in prediction, but rather weights them according to their residual.

**Extension to general quadratic minimization.** It is interesting to ask if this scheme can be used to minimize more general classes of quadratic problems. For example, in kernel SVM, the dual problem mimics a quadratic problem, and the kernel matrix is sparse, where nonzeros indicate training samples that are similar. By re-assigning the diagonal values, the kernel matrix becomes a (dense and signed) graph Laplacian – however, there are very few values that are very large (indicating similar training samples).

However, the question of whether this property holds *regardless* of diagonal rescaling is an open one. We find in practice that monotonicity does not hold, but the property that $\mathbf{supp}(\mathbf{x}^{(t)}) \subset \mathbf{supp}(\mathbf{x}^*)$ often does, in practice. Showing this theoretically would be interesting and powerful, but does not seem straightforward.

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

## A  Proofs from Section 2

**Lemma A.1** (Monotonicity and conservation (Andersen et al., 2006))**.** *Consider*

$$z = \frac{1+\alpha}{2\alpha} D^{1/2}(b - Qx).$$

*For all $t$, $\pi^{(t)} \geq 0$, $z^{(t)} \geq 0$. Moreover,*

$$\|z^{(t+1)}\|_1 \leq \|z^{(t)}\|_1, \qquad \|z^{(t)}\|_1 + \|D^{1/2}x^{(t)}\|_1 = 1.$$

*Proof.* First, it is clear that $z^{(t)} \geq 0$ for all $t$, since $z^{(0)} = e_s \geq 0$, and each subsequent operation either adds or scales positively. Then, it is also clear that $x^{(t)} \geq 0$, and therefore $\pi^{(t)} \geq 0$, and moreover that $\|D^{1/2}x^{(t)}\|_1 = \|\pi^{(t)}\|_1$ is monotonically increasing. To see that their weights are conserved, note that

$$\|z^{(t+1)}\|_1 - \|z^{(t)}\|_1 = \sum_i z_i^{(t+1)} - z_i^{(t)} = \sum_{v \in \mathcal{N}(u)} \frac{(1-\alpha)}{2d_u} z_u^{(t)} - \frac{(1+\alpha)}{2} z_u^{(t)} = -\alpha z_u^{(t)}$$

Therefore,

$$\|D^{1/2}x^{(t+1)}\|_1 - \|D^{1/2}x^{(t)}\|_1 = \|z^{(t)}\|_1 - \|z^{(t+1)}\|_1$$

and their mass is conserved. $\qquad\square$

### A.1  Extension to offline sparsification

The offline sparsification method essentially produces a new graph for which learning occurs. However, the original APPR method operates on the new sparsified graph exactly as it would over the original one, so the same properties of monotonicity and conservation is preserved.

### A.2  Extension to online sparsification (RandomAPPR)

**Lemma A.2** (Unbiased estimators)**.** *Define $\bar{z}^{(t)} = \frac{1+\alpha}{2\alpha} D^{1/2}(b - Qx^{(t)})$, $\bar{z}^{(t,i)} = \frac{1+\alpha}{2\alpha} D^{1/2}(b - Qx^{(t,i)})$. Then*

$$\mathbb{E}[\tilde{z}^{(t)}|x^{(t-1)}] = \bar{z}^{(t)}, \qquad \mathbb{E}[\tilde{z}^{(t,i)}|x^{(t-1)}] = \mathbb{E}[\bar{z}^{(t,i)}|x^{(t-1)}],$$

*Proof.* The first part of the claim is true by construction of $\tilde{z}^{(t)}$. Now, inductively, if

$$\mathbb{E}[\tilde{z}^{(t,i)}|x^{(t-1)}] = \mathbb{E}[\bar{z}^{(t,i)}|x^{(t-1)}]$$

then since

$$\begin{aligned}
\mathbb{E}[\tilde{z}^{(t,i+1)}|\tilde{z}_{u_i}^{(t,i)}] &= \tilde{z}^{(t,i)} + \tilde{z}_{u_i}^{(t,i)}\left(\frac{1-\alpha}{2}AD^{-1} - \frac{1+\alpha}{2}I\right)e_{u_i} \\
&= \tilde{z}^{(t,i)} - \frac{1+\alpha}{2}\tilde{z}_{u_i}^{(t,i)}D^{1/2}QD^{-1/2}e_{u_i}
\end{aligned}$$

therefore

$$\begin{aligned}
\mathbb{E}[\tilde{z}^{(t,i+1)}|x^{(t-1)}] &= \mathbb{E}[\tilde{z}^{(t,i)} - \frac{1+\alpha}{2}\tilde{z}_{u_i}^{(t,i)}D^{1/2}QD^{-1/2}e_{u_i}|x^{(t-1)}] \\
&= \mathbb{E}[\bar{z}^{(t,i)} - \frac{1+\alpha}{2}\bar{z}_{u_i}^{(t,i)}D^{1/2}QD^{-1/2}e_{u_i}|x^{(t-1)}]
\end{aligned}$$

At the same time

$$\begin{aligned}
\bar{z}^{(t,i+1)} &= \frac{1+\alpha}{2\alpha}D^{1/2}(b - Qx^{(t,i+1)}) \\
&= \bar{z}^{(t,i)} - \frac{1+\alpha}{2\alpha}D^{1/2}Q(x^{(t,i+1)} - x^{(t,i)}) \\
&= \bar{z}^{(t,i)} - \frac{1+\alpha}{2}D^{1/2}QD^{-1/2}(\tilde{z}_{u_i}^{(t,i)}e_{u_i})
\end{aligned}$$

and thus

$$\mathbb{E}[\bar{z}^{(t,i+1)}|\boldsymbol{x}^{(t-1)}] = \mathbb{E}[\bar{z}^{(t,i)} - \tfrac{1+\alpha}{2}\boldsymbol{D}^{1/2}\boldsymbol{Q}\boldsymbol{D}^{-1/2}(\tilde{z}_{u_i}^{(t,i)}\boldsymbol{e}_{u_i})|\boldsymbol{x}^{(t-1)}] = \mathbb{E}[\bar{z}^{(t,i)} - \tfrac{1+\alpha}{2}\bar{z}_{u_i}^{(t,i)}\boldsymbol{D}^{1/2}\boldsymbol{Q}\boldsymbol{D}^{-1/2}\boldsymbol{e}_{u_i}|\boldsymbol{x}^{(t-1)}]$$

$\square$

**Theorem A.3.** *In online APPR, for all $t$, $i \in \mathcal{S}_t$,*

$$\mathbb{E}[\bar{z}^{(t,i)}] \geq 0, \qquad \mathbb{E}[\bar{z}^{(t,i+1)}] \leq \mathbb{E}[\bar{z}^{(t,i)}], \qquad \mathbb{E}[\bar{z}^{(t+1)}] \leq \mathbb{E}[\bar{z}^{(t)}] \qquad \text{(monotonicity)}$$

*Furthermore,*

$$\mathbb{E}[\boldsymbol{x}^{(t,i)}] \geq 0, \qquad \mathbb{E}[\boldsymbol{x}^{(t,i+1)}] \geq \mathbb{E}[\boldsymbol{x}^{(t,i)}], \qquad \mathbb{E}[\boldsymbol{x}^{(t+1)}] \geq \mathbb{E}[\bar{\boldsymbol{x}}^{(t)}] \qquad \text{(monotonicity)}$$

*Moreover,*

$$\|\mathbb{E}[\boldsymbol{D}^{1/2}\boldsymbol{x}^{(t)}]\|_1 + \|\mathbb{E}[\tilde{z}^{(t)}]\|_1 = \|\boldsymbol{D}^{-1/2}\boldsymbol{b}\|_1 \qquad \text{(conservation)}$$

*and*

$$\|\mathbb{E}[\boldsymbol{D}^{1/2}\boldsymbol{x}^{(t+1)}]\|_1 - \|\mathbb{E}[\boldsymbol{D}^{1/2}\boldsymbol{x}^{(t)}]\|_1 = \|\mathbb{E}[\tilde{z}^{(t)}]\|_1 + \|\mathbb{E}[\tilde{z}^{(t+1)}]\|_1 \leq |\mathcal{S}_t|\alpha\epsilon \qquad \text{(descent)}$$

*Proof.* **Monotonicity of $\bar{r}$**

Clearly $\bar{z}^{(0)} = \tfrac{1+\alpha}{2\alpha}\boldsymbol{D}^{1/2}\boldsymbol{b} \geq 0$.

Now, if $\mathbb{E}[\bar{z}^{(t,i)}|\boldsymbol{x}^{(t-1)}] \geq 0$, then

$$\mathbb{E}[\bar{z}^{(t,i+1)}|\boldsymbol{x}^{(t-1)}] = \mathbb{E}[\bar{z}^{(t,i)}|\boldsymbol{x}^{(t-1)}] - \tfrac{1+\alpha}{2}\boldsymbol{D}^{1/2}\boldsymbol{Q}\boldsymbol{D}^{-1/2}(\mathbb{E}[\bar{z}_{u_i}^{(t,i)}|\boldsymbol{x}^{(t-1)}]\boldsymbol{e}_{u_i}) \geq 0.$$

The reasoning is the same as in Lemma A.1. Since this is true for all $\boldsymbol{x}^{(t-1)}$,

$$\mathbb{E}[\bar{z}^{(t,i+1)}] = \mathbb{E}[\bar{z}^{(t,i)}] - \tfrac{1+\alpha}{2}\boldsymbol{D}^{1/2}\boldsymbol{Q}\boldsymbol{D}^{-1/2}(\mathbb{E}[\bar{z}_{u_i}^{(t,i)}]\boldsymbol{e}_{u_i}) \geq 0.$$

Moreover, since $\mathbb{E}[\bar{z}_{u_i}^{(t,i)}] \geq 0$, so $\mathbb{E}[\boldsymbol{z}^{(t,i)}]$ is monotonically decreasing

Finally, since

$$\mathbb{E}[\bar{z}^{(t,|\mathcal{S}_t|)}] = \mathbb{E}[\mathbb{E}[\bar{z}^{(t,|\mathcal{S}_t|)}|\boldsymbol{x}^{(t-1)}]] = \mathbb{E}[\mathbb{E}[\bar{z}^{(t+1)}|\boldsymbol{x}^{(t-1)}]] = \mathbb{E}[\bar{z}^{(t+1)}]$$

these two properties hold for all $t$, $i$.

**Monotonicity of $x$**

Again, we begin with $\boldsymbol{x}^{(1)} = 0 \geq 0$. Then, since $\mathbb{E}[\tilde{z}^{(t,i)}] \geq 0$ for all $t, i$, then

$$\mathbb{E}[\boldsymbol{x}^{(t,i+1)}] = \mathbb{E}[\boldsymbol{x}^{(t,i)}] + \frac{\alpha}{\sqrt{d_{u_i}}}\mathbb{E}[\tilde{z}_{u_i}^{(t,i)}] \geq 0.$$

What's more, $\mathbb{E}[\boldsymbol{x}^{(t,i+1)}]$ is monotonically increasing.

**Conservation.**

$$\boldsymbol{1}^T\boldsymbol{D}^{1/2}(\boldsymbol{x}^{(t+1)} - \boldsymbol{x}^{(t)}) = \alpha\sum_{i\in\mathcal{S}_t}\tilde{z}_{u_i}^{(t,i)} = \boldsymbol{1}^T(\tfrac{1+\alpha}{2}\boldsymbol{I} - \tfrac{1-\alpha}{2}\boldsymbol{A}\boldsymbol{D}^{-1})\sum_{i\in\mathcal{S}_t}\tilde{z}_{u_i}^{(t,i)}\boldsymbol{e}_{u_i} = \boldsymbol{1}^T(\tilde{z}^{(t)} - \tilde{z}^{(t+1)})$$

so

$$\boldsymbol{1}^T\boldsymbol{D}^{1/2}\boldsymbol{x}^{(t)} = \boldsymbol{1}^T\boldsymbol{D}^{-1/2}\boldsymbol{e}_s - \boldsymbol{1}^T\tilde{z}^{(t)}$$

and since $\mathbb{E}[\boldsymbol{x}^{(t)}] \geq 0$ and $\mathbb{E}[\boldsymbol{z}^{(t)}] \geq 0$, then

$$\|\mathbb{E}[\boldsymbol{D}^{1/2}\boldsymbol{x}^{(t)}]\|_1 + \|\mathbb{E}[\tilde{z}^{(t)}]\|_1 = \|\boldsymbol{D}^{-1/2}\boldsymbol{b}\|_1$$

**Descent.** Finally, we can put it all together so that

$$\|\mathbb{E}[\boldsymbol{D}^{1/2}\boldsymbol{x}^{(t+1)}]\|_1 - \|\mathbb{E}[\boldsymbol{D}^{1/2}\boldsymbol{x}^{(t)}]\|_1 = \sum_{i \in \mathcal{S}_t} \alpha \mathbb{E}[\tilde{z}_{u_i}^{(t,i)}]$$

Now, for each $t, i$, it must be that $\mathbb{E}[\tilde{z}_{u_i}^{(t,i)}] \geq \boldsymbol{D}_{u_i, u_i} \epsilon$. This is because $u_i$ appears in $\mathcal{S}_t$ only once. So, either $\tilde{z}_{u_i}^{(t,i)} = \tilde{z}_u^{(t)}$ or a neighbor of $u_i$ pushed mass onto $\tilde{z}_{u_i}$. But, in the second case, mass can only be increased. So, in fact,

$$\|\mathbb{E}[\boldsymbol{D}^{1/2}\boldsymbol{x}^{(t+1)}]\|_1 - \|\mathbb{E}[\boldsymbol{D}^{1/2}\boldsymbol{x}^{(t)}]\|_1 \geq |\mathcal{S}_t| \alpha \epsilon$$

$\square$

# B  Concentration results for offline sparsification (Section 3)

## B.1  General facts about subgaussianity

**Definition.** A distribution with random variable $X$ is *subgaussian* with parameter $\sigma^2$ if

$$\mathbb{E}[\exp(\lambda(X - \mathbb{E}[X]))] \leq \exp(\sigma^2 \lambda^2 / 2), \quad \forall \lambda \in \mathbb{R}$$

[2]

We use the notation **subgauss**$(X)$ to say that a random variable $X$ is subgaussian with parameter **subgauss**$(X)$.

**Lemma B.1.** *Suppose $X$ is a Bernoulli random variable and $p = \mathbf{Pr}(X = 1) = 1 - \mathbf{Pr}(X = 0)$. Then*

$$\mathbf{subgauss}(X) \leq \min\{1/4, S_1(p)\}$$

*where*

$$S_1(p) := -\frac{p(p-b)}{2\ln(p)} + \frac{(p-b)^2}{4\ln^2(p)} \ln(p) + \frac{b(p-b)}{2\ln(p)}.$$

*Proof.* Consider

$$
\begin{aligned}
f(\lambda, p) := \frac{1}{\lambda^2} \ln(\mathbb{E}[\exp(\lambda(X - \mathbb{E}[X]))]) &= \frac{1}{\lambda^2} \ln(p \exp(\lambda(1-p)) + (1-p)\exp(\lambda(0-p))) \\
&= -\frac{p}{\lambda} + \frac{1}{\lambda^2} \ln(1 + p(\exp(\lambda) - 1))
\end{aligned}
$$

Then **subgauss**$(X)$ is any $\lambda$-independent upper bound on $f(\lambda, p)$.

First, we reframe the problem to $s = 1/\lambda$

$$g(s, p) = -ps + s^2 \ln(1 + p(\exp(1/s) - 1))$$

We show that if $s \geq 0$, then $g(-s, p) \leq g(s, p)$. This motivates that the maximum only occurs when $s \geq 0$. Specifically, if $s < 0$, then

$$s < 0 \Rightarrow \exp(1/s) < 1 \Rightarrow (1 - p + p \exp(1/s)) < 1 \Rightarrow \ln(1 - p + p \exp(1/s)) < 0.$$

So, if $s > 0$, then

$$g(s, p) - g(-s, p) = s^2 (\underbrace{\ln(1 - p + p\exp(1/s))}_{\geq 0} - \underbrace{\ln(1 - p + p\exp(-1/s))}_{\leq 0}) \geq 0.$$

---
[2]See also Wainwright (2019)

Next, we find $b$ such that for all $p < 0.5$ and $s$ satisfying

$$\ln(\frac{1-p}{p} + \exp(1/s)) \geq \frac{b}{s}$$

then

$$\frac{\partial g(s,p)}{\partial s} = -p + 2s\ln(1 + p(\exp(1/s) - 1)) - \frac{p\exp(1/s)}{1 + p(\exp(1/s) - 1)} < 0$$

e.g. as $s$ increases, $g(s,p)$ decreases. Numerically, we find $b$ very close to 1 is sufficient; we may thus take $b = 2$. Therefore,

$$\max_s g(s,p) \leq \max_s h(s,p) := -ps + s^2\ln(p) + bs$$

and it is sufficient to optimize over $h$. This is satisfied by

$$s = \frac{p - b}{2\ln(p)} \iff \lambda = \frac{2\ln(p)}{p - b}$$

and thus

$$\begin{aligned}
\mathbf{subgauss}(X) &\leq -\frac{p(p-b)}{2\ln(p)} + \frac{(p-b)^2}{4\ln^2(p)}\ln(p) + \frac{b(p-b)}{2\ln(p)} \\
&= -\frac{(p-b)^2}{4\ln(p)} \\
&=: S_1(p)
\end{aligned}$$

We can also include the bound through Hoeffding's lemma of $\mathbf{subgauss}(X) \leq 1/4$ to get

$$\mathbf{subgauss}(X) = S(p) := \begin{cases} 1/4 & 1/4 < S_1(p), 0 \leq p \leq 1 \\ S_1(p) & 1/4 \geq S_1(p), 0 \leq p \leq 1 \\ 1 & p > 1 \end{cases}$$

$\square$

A figure of the previous bound is shown in Figure 8

Figure 8: Bound on subgaussian constant for Bernoulli with parameter $p$.

**Lemma B.2.** *Suppose* $Z = \sum_{i=1}^{p} c_i X_i$ *where* $X_i$ *are Bernoulli random variables,* $\mathbf{Pr}(X_i = 1) = p_i$ *and* $\mathbf{Pr}(X_i = 0) = 1 - p_i$. *Then* $\mathbf{subgauss}(Z) = \sum_i c_i^2 S(p_i)$.

*Proof.* A Bernoulli random variable $X$ whose value can only be 0 or 1 is sub-Gaussian with parameter $\sigma^2 = S(p)$. Next, suppose that $X_1, X_2, ..., X_p$ are all subgaussian with parameters $\sigma_1^2, ..., \sigma_p^2$. Consider $Z = \sum_{i=1}^{p} c_i X_i$. Then

$$\mathbb{E}[\exp(\lambda \sum_i c_i x_i - \mu_i))] = \prod_{i=1}^{p} \mathbb{E}[\exp(\lambda c_i x_i - \lambda \mu_i)] \leq \prod_{i=1}^{p} \exp(\lambda^2 c_i^2 \sigma_i^2),$$

Therefore, $Z = \sum_{i=1}^p c_i X_i$ is subgaussian with parameter

$$\textbf{subgauss}(Z) \le \sum_i c_i^2 \textbf{subgauss}(X_i) \le \sum_i c_i^2 S(p_i).$$

$\square$

**Lemma B.3** (Chain rule)**.**

$$\textbf{subgauss}(X) \le \mathbb{E}[\textbf{subgauss}(X|Y)] + \textbf{subgauss}(\mathbb{E}[X|Y])$$

*Proof.* First, note that

$$\mathbb{E}[\exp(\lambda(X - \mathbb{E}[X]))] = \mathbb{E}[\mathbb{E}_{X|Y}[\exp(\lambda(X - \mathbb{E}[X]))|Y]] =$$

and

$$\mathbb{E}_{X|Y}[\exp(\lambda(X - \mathbb{E}[X]))|Y] = \mathbb{E}_U[\exp(\lambda(U - \mathbb{E}_Y[\mathbb{E}[U]]))]$$

for $U = X|Y$ (random in $X$ and a function in $Y$). Next, we may write

$$
\begin{aligned}
\mathbb{E}_U[\exp(\lambda(U - \mathbb{E}_Y[\mathbb{E}[U]]))] &= \mathbb{E}_U[\exp(\lambda(U - \mathbb{E}[U] + \mathbb{E}[U] - \mathbb{E}_Y[\mathbb{E}[U]]))] \\
&= \mathbb{E}_U[\exp(\lambda(U - \mathbb{E}[U])\exp(\lambda(\mathbb{E}[U] - \mathbb{E}_Y[\mathbb{E}[U]]))] \\
&\le \mathbb{E}_U[\exp(\lambda(U - \mathbb{E}[U])]\mathbb{E}_U[\exp(\lambda(\mathbb{E}[U] - \mathbb{E}_Y[\mathbb{E}[U]]))]
\end{aligned}
$$

where the inequality is from Cauchy Schwartz inequality. Similarly,

$$\mathbb{E}_Y[\mathbb{E}_U[\exp(\lambda(U - \mathbb{E}_Y[\mathbb{E}[U]]))]] \le \mathbb{E}_Y[\mathbb{E}_U[\exp(\lambda(U - \mathbb{E}[U]))]]\,\mathbb{E}_Y[\mathbb{E}_U[\exp(\lambda(\mathbb{E}[U] - \mathbb{E}_Y[\mathbb{E}[U]]))]]$$

Now, note that if $\mathbb{E}_Y[\textbf{subgauss}(U)] = \sigma_1$ and $\textbf{subgauss}(\mathbb{E}_U[U]) = \sigma_2$ then

$$\mathbb{E}_Y[\mathbb{E}_U[\exp(\lambda(U - \mathbb{E}[U]))]] \le \exp(\lambda^2 \sigma_1^2 / 2), \quad \mathbb{E}_Y[\exp(\lambda(\mathbb{E}_U[U] - \mathbb{E}_Y[\mathbb{E}_U[U]]))]] \le \exp(\lambda^2 \sigma_2^2 / 2),$$

which shows that

$$\mathbb{E}_Y[\mathbb{E}_U[\exp(\lambda(U - \mathbb{E}_Y[\mathbb{E}[U]]))]] \le \exp(\lambda^2(\sigma_1^2 + \sigma_2^2)/2).$$

$\square$

**Notation**   For a random vector $\boldsymbol{x}$, we write $\textbf{subgauss}(\boldsymbol{x}) = \max_i \textbf{subgauss}(\boldsymbol{x}_i)$.

## B.2   Offline sparsification

**Lemma B.4** (Subgaussianity of influencer sparsification)**.** *This extra result derives the subgaussian bound for this particular subsampling problem. However, it is not essential for proving our main result for offline sparsification.*

*Consider* $\boldsymbol{L} = \boldsymbol{I} - \boldsymbol{D}^{-1/2}\boldsymbol{A}\boldsymbol{D}^{-1/2}$, $\tilde{\boldsymbol{L}}$ *the Laplacian matrices corresponding to a graph and its sparsified version. Then, for any* $\boldsymbol{x} \in \mathbb{R}^n$,

$$\textbf{subgauss}(\boldsymbol{x}^T \tilde{\boldsymbol{L}}_I \boldsymbol{x}) = S(\frac{\bar{q}}{d_{\max}})\frac{d_{\max}^2}{4\bar{q}^2}\|\boldsymbol{x}\|_\infty^4 |\mathcal{S}_I|, \qquad \textbf{var}(\boldsymbol{x}^T \tilde{\boldsymbol{L}}_I \boldsymbol{x}) = \frac{\bar{q}}{\max\{d_i, d_j\}} \cdot (1 - \frac{\bar{q}}{\max\{d_i, d_j\}})\|\boldsymbol{x}\|_\infty^4 |\mathcal{S}_I|$$

*Proof.*

Denote $d_i$ as the degree of node $i$. We first separate the influencer rows and columns from $\boldsymbol{L}$, as

$$\mathcal{S}_I = \{(i,j) : d_i \geq \bar{q} \text{ or } d_j \geq \bar{q}\}$$

Now define $\boldsymbol{L}_C$ the parts of $\boldsymbol{L}$ that are not subsampled, e.g.

$$\boldsymbol{L}_C = \sum_{(i,j)\notin\mathcal{S}_I} \boldsymbol{L}_{i,j}\boldsymbol{e}_i\boldsymbol{e}_j^T$$

Define $\tilde{\boldsymbol{L}}_I = \tilde{\boldsymbol{L}} - \boldsymbol{L}_C$, $\boldsymbol{L}_I - \boldsymbol{L} - \boldsymbol{L}_C$.

Then, since $\boldsymbol{L}_C$ is not subsampled, $\mathbf{var}(\tilde{\boldsymbol{L}}) = \mathbf{var}(\tilde{\boldsymbol{L}}_I)$. and

$$\tilde{\boldsymbol{L}}_{i,j} = c_{i,j}\boldsymbol{L}_{i,j}$$

where

$$c_{i,j} = \begin{cases} \frac{\max\{d_i,d_j\}}{\bar{q}} & \text{w.p. } \frac{\bar{q}}{\max\{d_i,d_i\}} \\ 0 & \text{w.p. } 1 - \frac{\bar{q}}{\max\{d_i,d_i\}}, \end{cases} \quad (i,j) \in \mathcal{S}_I,$$

and 1 for $(i,j) \notin \mathcal{S}_I$. Then $\mathbb{E}[\tilde{\boldsymbol{L}}] = \boldsymbol{L}$ and $\boldsymbol{x}^T\tilde{\boldsymbol{L}}\boldsymbol{x}$ is subgaussian with parameter $\sigma^2$ if and only if $\boldsymbol{x}^T\tilde{\boldsymbol{L}}_I\boldsymbol{x}$ is subgaussian with parameter $\sigma^2$. Since $c_{i,j}$ is a scaled Bernoulli random variable which is subgaussian with parameter $\sigma_{i,j}^2 = \frac{S(p_{i,j})}{p_{i,j}^2} = S(\frac{\bar{q}}{\max\{d_i,d_j\}})\frac{\max\{d_i,d_j\}^2}{\bar{q}^2}$ and

$$\boldsymbol{x}^T\tilde{\boldsymbol{L}}_I\boldsymbol{x} = \sum_{(i,j)\in\mathcal{S}_I} (\boldsymbol{x}_i\boldsymbol{x}_j\boldsymbol{L}_{i,j})c_{i,j}$$

then $\boldsymbol{x}^T\tilde{\boldsymbol{L}}_I\boldsymbol{x}$ is subgaussian with parameter

$$\sum_{(i,j)\in\mathcal{S}_I} (\boldsymbol{x}_i\boldsymbol{x}_j\boldsymbol{L}_{i,j})^2 c_{i,j}^2 = \sum_{(i,j)\in\mathcal{S}_I} (\boldsymbol{x}_i\boldsymbol{x}_j\boldsymbol{L}_{i,j})^2 S(\frac{\bar{q}}{\max\{d_i,d_j\}})\frac{\max\{d_i,d_i\}^2}{\bar{q}^2}$$

$$\leq S(\frac{\bar{q}}{\max\{d_i,d_j\}})\frac{\max\{d_i,d_j\}^2}{\bar{q}^2} \sum_{(i,j)\in\mathcal{S}_I} (\boldsymbol{x}_i\boldsymbol{x}_j\boldsymbol{L}_{i,j})^2.$$

Similarly, taking the random variable $Z = \boldsymbol{x}^T\boldsymbol{L}_I\boldsymbol{x}$, then since $-1 \leq \boldsymbol{L}_{i,j} \leq 1$, then the following random variable is bounded

$$\mathbb{E}[Z] = \boldsymbol{x}^T\boldsymbol{L}_I\boldsymbol{x}, \qquad 0 \leq \boldsymbol{x}^T\boldsymbol{L}_I\boldsymbol{x} \leq \|\boldsymbol{x}\|_\infty^2|\mathcal{S}_I|.$$

and

$$\mathbf{var}(\boldsymbol{x}^T\boldsymbol{L}_I\boldsymbol{x}) = \sum_{(i,j)\in\mathcal{S}_I} (\boldsymbol{x}_i\boldsymbol{x}_j\boldsymbol{L}_{i,j})^2\mathbf{var}(c_{i,j})$$

$$= \sum_{(i,j)\in\mathcal{S}_I} (\boldsymbol{x}_i\boldsymbol{x}_j\boldsymbol{L}_{i,j})^2 \frac{\bar{q}}{\max\{d_i,d_j\}} \cdot (1 - \frac{\bar{q}}{\max\{d_i,d_j\}}) \leq \frac{1}{4} \sum_{(i,j)\in\mathcal{S}_I} (\boldsymbol{x}_i\boldsymbol{x}_j\boldsymbol{L}_{i,j})^2$$

Note also that $\boldsymbol{L}_{i,i} = 1$ and $|\boldsymbol{L}_{i,j}| \leq 1$, so

$$\sum_{(i,j)\in\mathcal{S}_I} (\boldsymbol{x}_i\boldsymbol{x}_j\boldsymbol{L}_{i,j})^2 \leq \sum_{(i,j)\in\mathcal{S}_I} (\boldsymbol{x}_i\boldsymbol{x}_j)^2 \leq \|\boldsymbol{x}\|_\infty^4|\mathcal{S}_I|$$

Overall, this yields is subgaussian with parameter

$$\mathbf{subgauss}(\boldsymbol{x}^T\tilde{\boldsymbol{L}}_I\boldsymbol{x}) = S(\frac{\bar{q}}{d_{\max}})\frac{d_{\max}^2}{\bar{q}^2}\|\boldsymbol{x}\|_\infty^4|\mathcal{S}_I|, \qquad \mathbf{var}(\boldsymbol{x}^T\tilde{\boldsymbol{L}}_I\boldsymbol{x}) = \frac{\|\boldsymbol{x}\|_\infty^4}{4}|\mathcal{S}_I|$$

$\square$

**Theorem B.5** (Offline sparsification). *The following holds pointwise over all* $\boldsymbol{x}$

$$\mathbf{Pr}(|\boldsymbol{x}^T \tilde{\boldsymbol{L}}_I \boldsymbol{x} - \boldsymbol{x}^T \boldsymbol{L}_I \boldsymbol{x}| \geq \epsilon) \leq 2 \min\{e^{-\frac{\epsilon^2}{8\|\boldsymbol{x}\|_\infty^2 \|\boldsymbol{x}\|_2^2}}, e^{-\frac{\epsilon^2}{8\|\boldsymbol{x}\|_\infty^4 |\mathcal{S}_I|}}\}.$$

*Proof.* The first two results are direct applications of Bernstein's inequality. The last result is proven below. Suppose $Z \sim \mathbf{Bern}(0, R)$ with mean $\mu$. Then for all $0 \leq \theta \leq 1$,

$$\mathbb{E}[e^{\lambda Z}] \overset{\text{convexity}}{\leq} (1 - \theta) + \theta e^{\lambda R} \overset{\theta = \mu/R}{=} 1 + (\mu/R)(e^{\lambda R} - 1)$$

$$\overset{1+x \leq e^x}{\leq} e^{(\mu/R)(e^{\lambda R} - 1)} \overset{e^x \leq 1 + x + x^2, 0 < x \leq 1}{\leq} e^{\mu\lambda + \mu\lambda^2 R}, \quad \lambda R \leq 1$$

Then, since $|\boldsymbol{x}_i \boldsymbol{x}_j \boldsymbol{L}_{i,j}| \leq \|\boldsymbol{x}\|_\infty^2$,

$$\mathbb{E}[e^{\lambda \boldsymbol{x}^T \tilde{\boldsymbol{L}}_I \boldsymbol{x}}] = \mathbb{E}[e^{\lambda \sum_{(i,j) \in \mathcal{S}_I} \boldsymbol{x}_i \boldsymbol{x}_j (\tilde{\boldsymbol{L}}_I)_{i,j}}]$$

$$\leq e^{\lambda \sum_{(i,j) \in \mathcal{S}_I} \boldsymbol{x}_i \boldsymbol{x}_j \boldsymbol{L}_{i,j}(1 + \lambda\|\boldsymbol{x}\|_\infty^2)} = e^{\lambda \boldsymbol{x}^T \boldsymbol{L}_I \boldsymbol{x}(1 + \lambda\|\boldsymbol{x}\|_\infty^2)}, \qquad \lambda\|\boldsymbol{x}\|_\infty^2 \leq 1.$$

By Chernoff's inequality,

$$\mathbf{Pr}(\boldsymbol{x}^T \tilde{\boldsymbol{L}}_I \boldsymbol{x} \geq (1 + \epsilon)\boldsymbol{x}^T \boldsymbol{L}_I \boldsymbol{x}) \quad \leq \quad \frac{\mathbb{E}[e^{\lambda \boldsymbol{x}^T \tilde{\boldsymbol{L}}_I \boldsymbol{x}}]}{e^{\lambda(1+\epsilon)\boldsymbol{x}^T \boldsymbol{L}_I \boldsymbol{x}}} \quad \forall \lambda > 0$$

$$\leq \quad e^{\lambda \boldsymbol{x}^T \boldsymbol{L}_I \boldsymbol{x}(\lambda\|\boldsymbol{x}\|_\infty^2 - \epsilon)} \quad \forall 0 < \lambda \leq \frac{1}{\|\boldsymbol{x}\|_\infty^2}$$

$$= \quad e^{-\frac{\boldsymbol{x}^T \boldsymbol{L}_I \boldsymbol{x}}{4\|\boldsymbol{x}\|_\infty^2}\epsilon^2}, \quad \lambda = \frac{\epsilon}{2\|\boldsymbol{x}\|_\infty^2}$$

Extending to a two-sided bound and using $\tilde{\epsilon} = \epsilon \boldsymbol{x}^T \boldsymbol{L}_I \boldsymbol{x}$,

$$\mathbf{Pr}(|\boldsymbol{x}^T \tilde{\boldsymbol{L}}_I \boldsymbol{x} - \boldsymbol{x}^T \boldsymbol{L}_I \boldsymbol{x}| \geq \tilde{\epsilon}) \leq 2 e^{-\frac{\tilde{\epsilon}^2}{4\|\boldsymbol{x}\|_\infty^2 \boldsymbol{x}^T \boldsymbol{L}_I \boldsymbol{x}}} \leq 2 \min\{e^{-\frac{\epsilon^2}{8\|\boldsymbol{x}\|_\infty^2 \|\boldsymbol{x}\|_2^2}}, e^{-\frac{\epsilon^2}{8\|\boldsymbol{x}\|_\infty^4 |\mathcal{S}_I|}}\}.$$

since in the last inequality, $\boldsymbol{x}^T \boldsymbol{L}_I \boldsymbol{x} \leq \|\boldsymbol{x}\|_2^2 \|\boldsymbol{L}_I\|_2 \leq \|\boldsymbol{x}\|_2^2$ or also $\boldsymbol{x}^T \boldsymbol{L}_I \boldsymbol{x} \leq \|\boldsymbol{x}\|_\infty^2 |\mathcal{S}_I|$.

$\square$

**Lemma B.6.** *For* $\boldsymbol{x} \sim \mathcal{N}(\boldsymbol{0}, \boldsymbol{I}_n)$, *for* $n > 3$,

$$\mathbf{Pr}(\frac{\|\boldsymbol{x}\|_\infty}{\|\boldsymbol{x}\|_2} \geq \sqrt{\frac{\log(n)}{n}}) \leq (n+1)\exp(-\frac{n\log(n)}{2})$$

*Proof.* From Vershynin (2018), for a Gaussian random variable $x$,

$$\mathbf{Pr}(x \geq \epsilon) \leq \frac{1}{\sqrt{2\pi}}\exp(-\frac{\epsilon^2}{2}), \qquad \epsilon \geq 1.$$

Applying union bound,

$$\mathbf{Pr}(\|\boldsymbol{x}\|_\infty \geq t_1) \leq \frac{n}{\sqrt{2\pi}}\exp(-\frac{t_1^2}{2}), \qquad t_1 \geq 1.$$

Also from Vershynin (2018),

$$\mathbf{Pr}(|\|x\|_2^2 - n| \geq \epsilon) \leq 2\exp(-\frac{1}{2}\min(\frac{\epsilon^2}{n}, \frac{\epsilon}{2}))$$

so picking $t_2^2 = n + \epsilon$,

$$\Pr\left(\|\boldsymbol{x}\|_2^2 \leq t_2^2\right) \leq 2\exp(-\frac{(n - t_2^2)^2}{2n})$$

Hence,

$$\mathbf{Pr}(\frac{\|\boldsymbol{x}\|_\infty}{\|\boldsymbol{x}\|_2} \leq \frac{t_1}{t_2}) \geq \mathbf{Pr}(\|\boldsymbol{x}\|_\infty \leq t_1 \text{ and } \|\boldsymbol{x}\|_2 \geq t_2)$$

$$= 1 - \mathbf{Pr}(\|\boldsymbol{x}\|_\infty \geq t_1 \text{ or } \|\boldsymbol{x}\|_2 \leq t_2) \geq 1 - \frac{n}{\sqrt{2\pi}}\exp(-\frac{t_1^2}{2}) - 2\exp(-\frac{(n - t_2^2)^2}{2n})$$

so,

$$\mathbf{Pr}(\frac{\|\boldsymbol{x}\|_\infty}{\|\boldsymbol{x}\|_2} \geq \frac{t_1}{t_2}) \leq \frac{n}{\sqrt{2\pi}}\exp(-\frac{t_1^2}{2}) + 2\exp(-\frac{(n - t_2^2)^2}{2n}).$$

Taking $t_2 = n$ and $t_1^2 = n\log(n)$ yields

$$\mathbf{Pr}(\frac{\|\boldsymbol{x}\|_\infty}{\|\boldsymbol{x}\|_2} \geq \sqrt{\frac{\log(n)}{n}}) \leq \frac{n}{\sqrt{2\pi}}\exp(-\frac{n\log(n)}{2}) + 2\exp(-\frac{(n - n^2)^2}{2n}) \leq (n+1)\exp(-\frac{n\log(n)}{2})$$

for $n > 3$. $\qquad\square$

**Lemma B.7** ( Rudelson & Vershynin (2013) Thm. 2.1). *Let* $\boldsymbol{x}_1, \ldots, \boldsymbol{x}_k \sim \mathcal{N}(0, \boldsymbol{I}_n)$ *be i.i.d. Gaussian vectors and* $\boldsymbol{A} \in \mathbb{R}^{n \times n}$. *Then*

$$\mathbf{Pr}(|\|\boldsymbol{A}\boldsymbol{x}\|_2 - \|\boldsymbol{A}\|_F| > \epsilon) \leq 2\exp(-\frac{c\epsilon^2}{s^4\|\boldsymbol{A}\|_2^2})$$

*where* $c$ *is a constant that does not depend on* $\boldsymbol{A}$ *or* $n$, *and* $s$ *is the subgaussian constant of a Gaussian random variable.*

This is a consequence of the Hanson-Wright inequality.

**Corollary B.7.1.** *For* $n \geq \max\{3, (\frac{32}{\epsilon^2}\ln(3))^{\frac{1}{2.75}}\}$,

$$\mathbf{Pr}(\boldsymbol{L}_I - \epsilon\boldsymbol{I} \preceq \tilde{\boldsymbol{L}}_I \preceq \boldsymbol{L}_I + \epsilon\boldsymbol{I}) \quad \geq \quad 1 - 12\exp(-\frac{\epsilon^2 n^3}{32\log(n)}) - 4\exp(-c'n\epsilon/2)$$

*Proof.* Using Thm. B.5, we can construct a pointwise conditional probability

$$\mathbf{Pr}\left(|\boldsymbol{x}^T\tilde{\boldsymbol{L}}_I\boldsymbol{x} - \boldsymbol{x}^T\boldsymbol{L}_I\boldsymbol{x}| \geq n\epsilon \,\middle|\, \frac{\|\boldsymbol{x}\|_\infty}{\|\boldsymbol{x}\|_2} \leq \sqrt{\frac{\log(n)}{n}}\right) \leq 2\exp(-\frac{\epsilon^2 n^3}{8\|\boldsymbol{x}\|_2^4\log(n)}).$$

From Lemma B.6, we also have that for a unit Gaussian vector $\boldsymbol{x} \in \mathbb{R}^n$, for $n > 3$

$$\mathbf{Pr}\left(\frac{\|\boldsymbol{x}\|_\infty}{\|\boldsymbol{x}\|_2} \geq \sqrt{\frac{\log(n)}{n}}\right) \leq (n+1)\exp(-\frac{n\log(n)}{2}) \leq 2n\exp(-\frac{n}{2}).$$

So,

$$\mathbf{Pr}\left(|\boldsymbol{x}^T\tilde{\boldsymbol{L}}_I\boldsymbol{x} - \boldsymbol{x}^T\boldsymbol{L}_I\boldsymbol{x}| \geq \epsilon\right) \quad \leq \quad \underbrace{\mathbf{Pr}\left(|\boldsymbol{x}^T\tilde{\boldsymbol{L}}_I\boldsymbol{x} - \boldsymbol{x}^T\boldsymbol{L}_I\boldsymbol{x}| \geq \epsilon \,\middle|\, \frac{\|\boldsymbol{x}\|_\infty}{\|\boldsymbol{x}\|_2} \leq \sqrt{\frac{\log(n)}{n}}\right)\mathbf{Pr}\left(\frac{\|\boldsymbol{x}\|_\infty}{\|\boldsymbol{x}\|_2} \leq \sqrt{\frac{\log(n)}{n}}\right)}_{2\exp(-\frac{\epsilon^2 n^3}{8\|\boldsymbol{x}\|_2^4\log(n)})}$$

$$+ \mathbf{Pr}\left(|\boldsymbol{x}^T\tilde{\boldsymbol{L}}_I\boldsymbol{x} - \boldsymbol{x}^T\boldsymbol{L}_I\boldsymbol{x}| \geq \epsilon \,\middle|\, \frac{\|\boldsymbol{x}\|_\infty}{\|\boldsymbol{x}\|_2} \geq \sqrt{\frac{\log(n)}{n}}\right)\underbrace{\mathbf{Pr}\left(\frac{\|\boldsymbol{x}\|_\infty}{\|\boldsymbol{x}\|_2} \geq \sqrt{\frac{\log(n)}{n}}\right)}_{\leq 2n\exp(-\frac{n}{2})}$$

$$\leq \quad 2\exp(-\frac{\epsilon^2 n}{8\|\boldsymbol{x}\|_2^4\log(n)}) + 2n\exp(-\frac{n}{2})).$$

Additionally,

$$
\begin{aligned}
\mathbf{Pr}\left(|\boldsymbol{x}^T \tilde{\boldsymbol{L}}_I \boldsymbol{x} - \boldsymbol{x}^T \boldsymbol{L}_I \boldsymbol{x}| \geq \epsilon\right) &= \mathbf{Pr}\left(|\boldsymbol{x}^T \tilde{\boldsymbol{L}}_I \boldsymbol{x} - \boldsymbol{x}^T \boldsymbol{L}_I \boldsymbol{x}| \geq \epsilon \ \middle| \ \|\boldsymbol{x}\|_2^2 \leq 2n\right) \mathbf{Pr}(\|\boldsymbol{x}\|_2^2 \leq 2n) \\
&\quad + \mathbf{Pr}\left(|\boldsymbol{x}^T \tilde{\boldsymbol{L}}_I \boldsymbol{x} - \boldsymbol{x}^T \boldsymbol{L}_I \boldsymbol{x}| \geq \epsilon \ \middle| \ \|\boldsymbol{x}\|_2^2 \geq 2n\right) \mathbf{Pr}(\|\boldsymbol{x}\|_2^2 \geq 2n) \\
&\leq \mathbf{Pr}\left(|\boldsymbol{x}^T \tilde{\boldsymbol{L}}_I \boldsymbol{x} - \boldsymbol{x}^T \boldsymbol{L}_I \boldsymbol{x}| \geq \epsilon \ \middle| \ \|\boldsymbol{x}\|_2^2 \leq 2n\right) + \mathbf{Pr}(\|\boldsymbol{x}\|_2^2 \geq 2n) \\
&\leq 2\exp(-\frac{\epsilon^2 n}{32\log(n)}) + 2n\exp(-\frac{n}{2})) + 2\exp(-\frac{n}{2}) \\
&\leq 12\exp(-\frac{\epsilon^2 n}{32\log(n)})
\end{aligned}
$$

And from Lemma B.7, taking $\boldsymbol{A} = \tilde{\boldsymbol{L}}_I - \boldsymbol{L}_I$ and using $\sqrt{n}\|\boldsymbol{A}\|_2 \geq \|\boldsymbol{A}\|_F$, for a single Gaussian random vector $\boldsymbol{x} \sim \mathcal{N}(0, \boldsymbol{I})$,

$$
\mathbf{Pr}\left(\|\tilde{\boldsymbol{L}}_I - \boldsymbol{L}_I\|_2 > \frac{\sqrt{\boldsymbol{x}^T(\tilde{\boldsymbol{L}}_I - \boldsymbol{L}_I)\boldsymbol{x}}}{\sqrt{n}} + \sqrt{\epsilon}\right) \leq 2\exp\left(-c'n\epsilon\right)
$$

where $c' = c/(2s^4)$. Squaring both sides and multiplying by $\sqrt{n}$,

$$
\begin{aligned}
2\exp\left(-c'n\epsilon\right) &\geq \mathbf{Pr}\left(n\|\tilde{\boldsymbol{L}}_I - \boldsymbol{L}_I\|_2^2 > \boldsymbol{x}_i^T(\tilde{\boldsymbol{L}}_I - \boldsymbol{L}_I)\boldsymbol{x}_i + n\epsilon + \sqrt{n\epsilon\boldsymbol{x}_i^T(\tilde{\boldsymbol{L}}_I - \boldsymbol{L}_I)\boldsymbol{x}_i},\right) \\
&= \mathbf{Pr}\left(n\|\tilde{\boldsymbol{L}}_I - \boldsymbol{L}_I\|_2^2 > \boldsymbol{x}_i^T(\tilde{\boldsymbol{L}}_I - \boldsymbol{L}_I)\boldsymbol{x}_i + 2n\epsilon, \ i = 1, ..., K\right) \mathbf{Pr}(\boldsymbol{x}_i^T(\tilde{\boldsymbol{L}}_I - \boldsymbol{L}_I)\boldsymbol{x}_i \leq n\epsilon) \\
&\quad + \mathbf{Pr}\left(n\|\tilde{\boldsymbol{L}}_I - \boldsymbol{L}_I\|_2^2 > \boldsymbol{x}^T(\tilde{\boldsymbol{L}}_I - \boldsymbol{L}_I)\boldsymbol{x} + n\epsilon + \sqrt{n\epsilon\boldsymbol{x}^T(\tilde{\boldsymbol{L}}_I - \boldsymbol{L}_I)\boldsymbol{x}} \ \middle| \ \frac{\boldsymbol{x}^T(\tilde{\boldsymbol{L}}_I - \boldsymbol{L}_I)\boldsymbol{x}}{n} \geq \epsilon)\right) \\
&\quad \cdot \mathbf{Pr}(\boldsymbol{x}^T(\tilde{\boldsymbol{L}}_I - \boldsymbol{L}_I)\boldsymbol{x} \geq n\epsilon) \\
&\geq \mathbf{Pr}\left(n\|\tilde{\boldsymbol{L}}_I - \boldsymbol{L}_I\|_2^2 > \boldsymbol{x}^T(\tilde{\boldsymbol{L}}_I - \boldsymbol{L}_I)\boldsymbol{x} + 2n\epsilon\right) \underbrace{\mathbf{Pr}(\boldsymbol{x}^T(\tilde{\boldsymbol{L}}_I - \boldsymbol{L}_I)\boldsymbol{x} \leq n\epsilon)}_{\geq 1 - 12\exp(-\frac{\epsilon^2 n^3}{32\log(n)})}
\end{aligned}
$$

For $n$ large enough,

$$
12\exp(-\frac{\epsilon^2 n^3}{32\log(n)}) \leq 12\exp(-\frac{\epsilon^2 n^{2.75}}{32}) \leq D \iff n \geq (\frac{32}{\epsilon^2}\ln(\frac{12}{D}))^{\frac{1}{2.75}}.
$$

Pick $D = 1/2$. Then in that regime,

$$
\mathbf{Pr}\left(n\|\tilde{\boldsymbol{L}}_I - \boldsymbol{L}_I\|_2^2 > \boldsymbol{x}^T(\tilde{\boldsymbol{L}}_I - \boldsymbol{L}_I)\boldsymbol{x} + n\epsilon\right) \leq 4\exp(-c'n\epsilon),
$$

So,

$$
\begin{aligned}
\mathbf{Pr}(\boldsymbol{L}_I - \epsilon\boldsymbol{I} \preceq \tilde{\boldsymbol{L}}_I \preceq \boldsymbol{L}_I + \epsilon\boldsymbol{I}) &= 1 - \mathbf{Pr}(n\|\boldsymbol{L}_I - \tilde{\boldsymbol{L}}_I\| \geq n\epsilon) \\
&\geq 1 - \mathbf{Pr}(n\|\boldsymbol{L}_I - \tilde{\boldsymbol{L}}_I\| > \boldsymbol{x}^T(\tilde{\boldsymbol{L}}_I - \boldsymbol{L}_I)\boldsymbol{x} + \epsilon \quad \text{or} \quad \boldsymbol{x}^T(\tilde{\boldsymbol{L}}_I - \boldsymbol{L}_I)\boldsymbol{x} \geq \epsilon) \\
&\geq 1 - 12\exp(-\frac{\epsilon^2 n^3}{32\log(n)}) - 4\exp\left(-c'n\epsilon/2\right)
\end{aligned}
$$

This is a uniform spectral bound. $\qquad\square$

## C  Convergence results for offline sparsification (Section 4)

**Assumption C.1.** *There exists constants $\sigma^{(t)}$ and $\sigma^{(t,i)}$ for $t = 1, ..., T$ and $i = 1, ..., |\mathcal{S}^{(t)}|$ such that*

- *the random variable $\tilde{z}_j^{(t)}|\boldsymbol{x}^{(t)}$ is subgaussian with parameter $(\sigma^{(t)})^2$, for all $j$*

- *the random variable $\tilde{z}_j^{(t,i+1)}|\tilde{\boldsymbol{z}}^{(t,i)}$ is subgaussian with parameter $(\sigma^{(t,i)})^2$, for all $j$*

**Assumption C.2.** *There exists a constant $R$ upper bounding each residual term*

$$\max\{\|\tilde{\boldsymbol{z}}^{(t)}\|_\infty, \|\tilde{\boldsymbol{z}}^{(t,i)}\|_\infty\} \leq R,$$

*for all $t = 1, ..., T, \ i = 1, ..., |\mathcal{S}_t|$.*

**Theorem C.1.** *Consider the online version of the algorithm. The probability that for some $i$, $\boldsymbol{D}^{-1}z_i^{(t)} > \epsilon$ but $\boldsymbol{D}^{-1}\tilde{z}_i^{(t)} < c\epsilon$ is bounded by*

$$\mathbf{Pr}(|\boldsymbol{D}^{-1}z_i^{(t)}| > \epsilon \text{ and } |\boldsymbol{D}^{-1}\tilde{z}_i^{(t)}| < c\epsilon) \leq \exp\left(-\frac{(1-c)^2\epsilon^2}{2\sigma_t^2}\right).$$

*Proof.* This is the result of a direct application of a Hoeffding bound:

$$\mathbf{Pr}(|\boldsymbol{D}^{-1}z_i^{(t)}| > \epsilon \text{ and } |\boldsymbol{D}^{-1}\tilde{z}_i^{(t)}| < c\epsilon) \leq \mathbf{Pr}(|\boldsymbol{D}^{-1}\tilde{z}_i^{(t)} - \boldsymbol{D}^{-1}z_i^{(t)}| < (1-c)\epsilon) \leq \exp\left(-\frac{(1-c)^2\epsilon^2}{2\sigma_t^2}\right)$$

$\square$

**Lemma C.2.** *Using SAMPLER, the constants*

$$(\sigma^{(t)})^2 \leq \frac{S(p_i)}{p_i^2}|\mathbf{supp}(\boldsymbol{x}^*)|, \qquad (\sigma^{(t,i)})^2 \leq \frac{S(p_i)}{p_i^2}$$

*Proof.* SAMPLER, forms

$$\tilde{\boldsymbol{w}}_i = \begin{cases} \frac{\boldsymbol{w}_i}{p_i}, & \text{w.p. } p_i \\ 0, & \text{else.} \end{cases}$$

then

$$\mathbf{subgauss}(\tilde{\boldsymbol{w}}_i) \leq (\frac{\boldsymbol{w}_i}{p_i})^2 S(p_i).$$

So, therefore,

$$(\sigma^{(t)})^2 = (\frac{\|\boldsymbol{x}\|_\infty}{p_i})^2 S(p_i)|\mathbf{supp}(\boldsymbol{x}^{(t)})| \leq \frac{S(p_i)R}{p_i^2}|\mathbf{supp}(\boldsymbol{x}^*)|,$$

$$(\sigma^{(t,i)})^2 = (\frac{\|\boldsymbol{z}^{(t,i)}\|_\infty}{p_i})^2 S(p_i) \leq \frac{S(p_i)R}{p_i^2}$$

since $R$ upper bounds the max norm of $\boldsymbol{z}^{(t,i)}$, which in turn bounds the mass that can be pused to $\boldsymbol{x}^{(t,i)}$.

$\square$

**Lemma C.3.** *Under assumptions C.2 and C.1, define $\tilde{\boldsymbol{Q}} = \frac{1+\alpha}{2}\boldsymbol{D}^{1/2}\boldsymbol{Q}\boldsymbol{D}^{-1/2}$. Then*

$$\mathbb{E}[\tilde{\boldsymbol{z}}^{(t,i+1)}|\tilde{\boldsymbol{z}}^{(t,i)}] = (\boldsymbol{I} - \tilde{\boldsymbol{Q}}\boldsymbol{e}_{s_i}\boldsymbol{e}_{s_i}^T)\tilde{\boldsymbol{z}}^{(t,i)}, \qquad \mathbb{E}[\tilde{\boldsymbol{z}}^{(t,i)}|\tilde{\boldsymbol{z}}^{(t)}] = \prod_{j \in \mathcal{S}_t}(\boldsymbol{I} - \tilde{\boldsymbol{Q}}\boldsymbol{e}_{s_j}\boldsymbol{e}_{s_j}^T)\tilde{\boldsymbol{z}}^{(t)}$$

$$\mathbf{subgauss}(\tilde{\boldsymbol{z}}_j^{(t,i+1)}) \leq (\sigma^{(t,i)})^2 + R_0\max_k \mathbf{subgauss}(\tilde{\boldsymbol{z}}_k^{(t,i)})$$

$$\mathbf{subgauss}(\tilde{\boldsymbol{z}}_j^{(t,i)}) \leq \sum_{j=1}^i R_0^{j-1}(\sigma^{(t,j)})^2 + R_0^i(\sigma^{(t)})^2.$$

*Proof.* We have already previously shown that

$$\mathbb{E}[\tilde{\boldsymbol{z}}^{(t,i+1)}|\tilde{\boldsymbol{z}}^{(t,i)}] = (\boldsymbol{I} - \tilde{\boldsymbol{Q}}\boldsymbol{e}_{s_i}\boldsymbol{e}_{s_i}^T)\tilde{\boldsymbol{z}}^{(t,i)}$$

so using chain rule,

$$\mathbb{E}[\tilde{\boldsymbol{z}}^{(t,i)}|\tilde{\boldsymbol{z}}^{(t)}] = \prod_{j \in \mathcal{S}_t} (\boldsymbol{I} - \tilde{\boldsymbol{Q}}\boldsymbol{e}_{s_j}\boldsymbol{e}_{s_j}^T)\tilde{\boldsymbol{z}}^{(t)}$$

Using Lemma B.3

$$
\begin{aligned}
\mathbf{subgauss}(\tilde{z}_j^{(t,i+1)}) &= \mathbb{E}[\mathbf{subgauss}(\tilde{z}_j^{(t,i+1)}|\tilde{z}_j^{(t,i)})] + \mathbf{subgauss}(\mathbb{E}[\tilde{\boldsymbol{z}}_j^{(t,i+1)}|\tilde{z}_j^{(t,i)}]) \\
&= (\sigma^{(t,i)})^2 + \mathbf{subgauss}(((\boldsymbol{I} - \tilde{\boldsymbol{Q}}\boldsymbol{e}_{s_i}\boldsymbol{e}_{s_i}^T)\tilde{\boldsymbol{z}}^{(t,i)})_j) \\
&\leq (\sigma^{(t,i)})^2 + R_0 \max_k \mathbf{subgauss}(\tilde{z}_k^{(t,i)})
\end{aligned}
$$

where $\|\boldsymbol{I} - \tilde{\boldsymbol{Q}}\boldsymbol{e}_{s_i}\boldsymbol{e}_{s_i}^T\|_\infty = 1$. Telescoping,

$$
\begin{aligned}
\mathbf{subgauss}(\tilde{z}_j^{(t,i)}) &\leq \sum_{j=1}^i R_0^{j-1}(\sigma^{(t,j)})^2 + R_0{}^i \max_k \mathbf{subgauss}(\tilde{z}_k^{(t)}) \\
&\leq \sum_{j=1}^i R_0^{j-1}(\sigma^{(t,j)})^2 + R_0^i(\sigma^{(t)})^2.
\end{aligned}
$$

$\square$

**Theorem C.4** (1/T rate). *Consider* $f(\boldsymbol{x}) = \frac{1}{2}\boldsymbol{x}^T\boldsymbol{Q}\boldsymbol{x} - \boldsymbol{b}^T\boldsymbol{x}$ *where* $\boldsymbol{b} = \frac{2\alpha}{1+\alpha}\boldsymbol{D}^{-1/2}\boldsymbol{e}_s$. *Initialize* $\boldsymbol{x}^{(0)} = 0$. *Define* $M^{(t)} = \sum_{\tau=1}^t |\mathcal{S}_t|$ *the number of push calls at epoch* $t$. *Then,*

$$\min_{t,j} \|\nabla f(\boldsymbol{x}^{(t,j)})\|_2^2 \leq \frac{1}{\alpha M^{(t)}} + \frac{\alpha \sigma_{\max}^2}{2}$$

*Here,* $\alpha$ *can be chosen to mitigate the tradeoff between convergence rate and final noise level.*

*Proof.*

$$
\begin{aligned}
\mathbb{E}[f(\boldsymbol{x}^{(t,i+1)})|\boldsymbol{x}^{(t,i)}] &\leq f(\boldsymbol{x}^{(t,i)}) + \mathbb{E}[\nabla f(\boldsymbol{x}^{(t,i)})^T(\boldsymbol{x}^{(t,i+1)} - \boldsymbol{x}^{(t,i)})|\boldsymbol{x}^{(t,i)}] + \mathbb{E}[\|\boldsymbol{x}^{(t,i+1)} - \boldsymbol{x}^{(t,i)}\|_2^2|\boldsymbol{x}^{(t)}] \\
&= f(\boldsymbol{x}^{(t,i)}) + \mathbb{E}[(\boldsymbol{Q}\boldsymbol{x}^{(t,i)} - \boldsymbol{b})^T(\boldsymbol{x}^{(t,i+1)} - \boldsymbol{x}^{(t,i)})|\boldsymbol{x}^{(t,i)}] + \mathbb{E}[\|\boldsymbol{x}^{(t,i+1)} - \boldsymbol{x}^{(t,i)}\|_2^2|\boldsymbol{x}^{(t,i)}] \\
&= f(\boldsymbol{x}^{(t,i)}) + (\boldsymbol{Q}\boldsymbol{x}^{(t,i)} - \boldsymbol{b})^T(\mathbb{E}[\boldsymbol{x}^{(t,i+1)}|\boldsymbol{x}^{(t,i)}] - \boldsymbol{x}^{(t,i)}) + \mathbb{E}[\|\boldsymbol{x}^{(t,i+1)} - \boldsymbol{x}^{(t,i)}\|_2^2|\boldsymbol{x}^{(t,i)}]
\end{aligned}
$$

Note that $\boldsymbol{x}^{(t,i+1)} - \boldsymbol{x}^{(t,i)} = \alpha\boldsymbol{D}^{-1/2}\tilde{\boldsymbol{z}}_{s_{i+1}}^{(t,i+1)}$, so

$$\mathbb{E}[\boldsymbol{x}^{(t,i+1)}|\boldsymbol{x}^{(t,i)}] = \boldsymbol{x}^{(t,i)} + \alpha\boldsymbol{D}^{-1/2}\mathbb{E}[\tilde{\boldsymbol{z}}_{s_{i+1}}^{(t,i+1)}|\boldsymbol{x}^{(t,i)}] = \boldsymbol{x}^{(t,i)} + \alpha\boldsymbol{D}^{-1/2}(\boldsymbol{b} - \boldsymbol{Q}\boldsymbol{x}^{(t,i)})_{e_{s_i}}$$

and for $d = \mathbf{diag}(\boldsymbol{D})$, $\min_i d_i \geq 1$,

$$
\begin{aligned}
\mathbb{E}[\|\boldsymbol{x}^{(t,i+1)} - \boldsymbol{x}^{(t,i)}\|_2^2|\boldsymbol{x}^{(t,i)}] &= \alpha^2\mathbb{E}[(d_{s_{i+1}}^{-1/2}\tilde{z}_{s_{i+1}}^{(t,i+1)})^2|\boldsymbol{x}^{(t,i)}] \\
&= \frac{\alpha^2}{d_{s_{i+1}}}\mathbf{var}(\tilde{z}_{s_{i+1}}^{(t,i+1)}|\boldsymbol{x}^{(t,i)}) - \frac{\alpha^2}{d_{s_{i+1}}}\mathbb{E}[\tilde{z}_{s_{i+1}}^{(t,i+1)}|\boldsymbol{x}^{(t,i)}]^2 \\
&\leq \alpha^2(\sigma^{(t,i+1)})^2
\end{aligned}
$$

so, using $\nabla f(\boldsymbol{x}) = \boldsymbol{Q}\boldsymbol{x} - \boldsymbol{b}$,

$$\mathbb{E}[f(\boldsymbol{x}^{(t,i+1)})|\boldsymbol{x}^{(t,i)}] \leq f(\boldsymbol{x}^{(t,i)}) - \alpha\|\nabla f(\boldsymbol{x}^{(t,i)})\|_2^2 + \frac{\alpha^2(\sigma^{(t,i+1)})^2}{2}$$

Telescoping over one epoch,

$$\mathbb{E}[f(\boldsymbol{x}^{(t,i+1)})|\boldsymbol{x}^{(t)}] - f(\boldsymbol{x}^{(t)}) \leq -\alpha\sum_{j=1}^{i+1}\|\nabla f(\boldsymbol{x}^{(t,j)})\|_2^2 + (i-1)\frac{\alpha^2\sigma_{\max}^2}{2}.$$

Since $\boldsymbol{x}^{(t+1)} = \boldsymbol{x}^{(t,|\mathcal{S}^{(t)}|)}$, telescope again

$$\mathbb{E}[f(\boldsymbol{x}^{(t)})] - f(\boldsymbol{x}^{(0)}) \leq -\alpha\sum_{\tau=1}^{t}\sum_{j=1}^{|\mathcal{S}^{(\tau)}|}\|\nabla f(\boldsymbol{x}^{(t,j)})\|_2^2 + \frac{M^{(t)}\alpha^2\sigma_{\max}^2}{2}$$

Then rearranging,

$$\frac{1}{M^{(t)}}\sum_{\tau=1}^{t}\sum_{j=1}^{|\mathcal{S}^{(\tau)}|}\|\nabla f(\boldsymbol{x}^{(t,j)})\|_2^2 \leq \frac{f(\boldsymbol{x}^{(0)}) - \mathbb{E}[f(\boldsymbol{x}^{(t)})]}{\alpha M^{(t)}} + \frac{\alpha\sigma_{\max}^2}{2} \leq \frac{f(\boldsymbol{x}^{(0)}) - f^*}{\alpha M^{(t)}} + \frac{\alpha\sigma_{\max}^2}{2}$$

One can pick $\alpha \in (0,1)$ to mitigate this tradeoff. $\qquad\square$

**Theorem C.5** (Linear rate in expectation). *Using $\bar{\boldsymbol{z}}^{(t)} = \frac{1+\alpha}{2\alpha}\boldsymbol{D}^{1/2}(\boldsymbol{b} - \boldsymbol{Q}\boldsymbol{x}^{(t)})$, in expectation,*

$$\|\mathbb{E}[\tilde{\boldsymbol{z}}^{(t+1)}]\|_1 \leq \exp\left(-\frac{M_t\alpha\epsilon}{R}\right).$$

*Moreover, since $\|\boldsymbol{z}^{(t)}\|_1 \leq \sqrt{n}\|\boldsymbol{z}^{(t)}\|_2$,*

$$\mathbf{Pr}\left(\|\boldsymbol{z}^{(t)}\|_1 \geq \exp\left(-\frac{M_t\alpha\epsilon}{R}\right)^{M_t} + \epsilon\right) \leq \exp\left(-\frac{\epsilon^2}{2\alpha^2\sqrt{n}\left(\sum_{\tau=1}^{t}\sum_{i\in\mathcal{S}_t}\sigma^{(t,i)} + \sum_{\tau=1}^{t}\sigma^{(t)}\right)}\right)$$

*Proof.* In Th. C.1 we already showed that

$$\|\mathbb{E}[\boldsymbol{D}^{1/2}\boldsymbol{x}^{(t+1)}]\|_1 - \|\mathbb{E}[\boldsymbol{D}^{1/2}\boldsymbol{x}^{(t)}]\|_1 \geq |\mathcal{S}_t|\alpha\epsilon$$

so by conservation,

$$\|\mathbb{E}[\tilde{\boldsymbol{z}}^{(t)}]\|_1 - \|\mathbb{E}[\tilde{\boldsymbol{z}}^{(t+1)}]\|_1 \geq |\mathcal{S}_t|\alpha\epsilon \geq |\mathcal{S}_t|\alpha\epsilon\frac{\|\mathbb{E}[\tilde{\boldsymbol{z}}^{(t)}]\|_1}{R}$$

where the last step uses the assumption that $\|\tilde{\boldsymbol{z}}^{(t)}\|_2 \leq R$.

Then

$$\|\mathbb{E}[\tilde{\boldsymbol{z}}^{(t+1)}]\|_1 \leq \|\mathbb{E}[\tilde{\boldsymbol{z}}^{(t)}]\|_1\left(1 - \frac{|\mathcal{S}_t|\alpha\epsilon}{R}\right) \leq \underbrace{\|\boldsymbol{e}_s\|_1}_{=1}\prod_{\tau=1}^{t}\left(1 - \frac{|\mathcal{S}_\tau|\alpha\epsilon}{R}\right) \leq \prod_{\tau=1}^{t}\exp\left(-\frac{|\mathcal{S}_\tau|\alpha\epsilon}{R}\right)$$

Writing $M_t = \sum_{\tau=1}^{t}|\mathcal{S}_\tau|$ gives

$$\|\mathbb{E}[\tilde{\boldsymbol{z}}^{(t+1)}]\|_1 \leq \exp\left(-\frac{M_t\alpha\epsilon}{R}\right).$$

Next, note that

$$\mathbf{subgauss}(\|\boldsymbol{D}^{1/2}\boldsymbol{x}^{(t)}\|_1) = \mathbf{subgauss}(\alpha \sum_{\tau=1}^{t} \sum_{i \in \mathcal{S}_t} \tilde{\boldsymbol{r}}^{(t,i)}) = \alpha^2 \sum_{\tau=1}^{t} \sum_{i \in \mathcal{S}_t} \sigma^{(t,i)} + \alpha^2 \sum_{\tau=1}^{t} \sigma^{(t)}$$

Applying Hoeffding's bound does the rest.

$\square$

## D   Online learning results (Section 5.1)

**Property 1.** If $\boldsymbol{y}$ is $\gamma$-smooth w.r.t. a graph with Laplacian $\boldsymbol{L}$, and a sparsified graph incurs $\boldsymbol{L}'$ where $|\boldsymbol{x}^T(\boldsymbol{L} - \boldsymbol{L}')\boldsymbol{x}| \leq \epsilon \boldsymbol{x}^T \boldsymbol{x}$, then $\boldsymbol{y}$ is $(\gamma + \epsilon n)$-smooth w.r.t. a graph with Laplacian $\boldsymbol{L}'$.

**Theorem D.1.** *RELAXATION using $\gamma' = \gamma + \epsilon n$ achieves*

$$\text{regret}(n) \leq \sqrt{\mathbf{tr}\left(\left(\frac{\boldsymbol{L}'}{2\gamma'} + \frac{\boldsymbol{I}}{2n}\right)^{-1}\right)} \leq D\sqrt{2n^{1+\rho}} + \sqrt{\frac{\epsilon n}{1 - \beta}}.$$

*Proof.*

$$
\begin{aligned}
\text{regret}_{\mathcal{G}}(n) - \text{regret}_{\tilde{\mathcal{G}}}(n) &= \sqrt{\mathbf{tr}\left(\left(\frac{\boldsymbol{L}'}{2\gamma'} + \frac{\boldsymbol{I}}{2n}\right)^{-1}\right)} - \sqrt{\mathbf{tr}\left(\left(\frac{\boldsymbol{L}}{2\gamma} + \frac{\boldsymbol{I}}{2n}\right)^{-1}\right)} \\
&\leq \sqrt{\mathbf{tr}\left(\left(\frac{\boldsymbol{L}'}{2\gamma'} + \frac{\boldsymbol{I}}{2n}\right)^{-1}\right) - \mathbf{tr}\left(\left(\frac{\boldsymbol{L}}{2\gamma} + \frac{\boldsymbol{I}}{2n}\right)^{-1}\right)} \\
&= \sqrt{\mathbf{tr}\left(\left(\frac{\boldsymbol{L}'}{2\gamma'} + \frac{\boldsymbol{I}}{2n}\right)^{-1} - \left(\frac{\boldsymbol{L}}{2\gamma} + \frac{\boldsymbol{I}}{2n}\right)^{-1}\right)} \\
&= \sqrt{\sum_{i=1}^{n} \frac{2n\gamma'}{\lambda_i' n + \gamma'} - \frac{2n\gamma}{\lambda_i n + \gamma}} \\
&\leq \sqrt{\sum_{i=1}^{n} \frac{2n(\gamma + \epsilon n)}{(\lambda_i - \epsilon)n + \gamma + \epsilon n} - \frac{2n\gamma}{\lambda_i n + \gamma}}
\end{aligned}
$$

Since

$$\frac{2n(\gamma + \epsilon n)}{(\lambda_i - \epsilon)n + \gamma + \epsilon n} - \frac{2n\gamma}{\lambda_i n + \gamma} = \frac{2n(\gamma + \epsilon n)}{\lambda_i n + \gamma} - \frac{2n\gamma}{\lambda_i n + \gamma} = \frac{\epsilon n}{\lambda_i n + \gamma} \leq \frac{\epsilon}{\lambda_{\min}}$$

then

$$\text{regret}_{\mathcal{G}}(n) - \text{regret}_{\tilde{\mathcal{G}}}(n) \leq \sqrt{\frac{\epsilon n}{\lambda_{min}}}$$

Picking a kernel such that $\lambda_{\min}(\boldsymbol{L}) = 1 - \beta$ yields the result.

$\square$

# E   Extended numerical section

**Data statistics.**   Table 1 gives a summary of the graph characteristics of several large graphs. Our experiments cover the first 7. We added a few very large graphs that we could not compute ourselves due to lack of computational resources, to highlight the heavy tailed degree distribution (by comparing the mean/median/max node degree values).

| | # nodes | # edges | avg. n.d. | median. n.d. | max n.d. | max n.d. / # nodes | max n.d. / avg. n.d. |
|---|---|---|---|---|---|---|---|
| Datasets in this paper | | | | | | | |
| political | 1222 | 33431 | 27.35 | 13 | 351 | 0.28 | 12.83 |
| citeseer | 2110 | 7336 | 3.47 | 2 | 99 | 0.046 | 28.47 |
| cora | 2485 | 10138 | 4.079 | 3 | 168 | 0.067 | 41.17 |
| pubmed | 19717 | 88648 | 4.49 | 2 | 171 | 0.0086 | 38.03 |
| mnist | 12000 | 194178 | 16.1815 | 12 | 226 | 0.018 | 13.96 |
| blogcatalog | 10312 | 667966 | 64.77 | 21 | 3992 | 0.38 | 61.62 |
| ogbn-arxiv | 169343 | 1166243 | 13.7 | 6 | 13161 | 0.077 | 960.65 |
| Other datasets | | | | | | | |
| facebook (artist) | 50515 | 819306 | 32.4 | 13 | 1469 | 0.029 | 45.339 |
| Amazon0302 | 262111 | 899792 | 6.86 | 6 | 420 | 0.0016 | 61.22 |
| com-dblp | 317080 | 1049865 | 6.62 | 4 | 343 | 0.0011 | 51.81 |
| web-Google | 875713 | 4322051 | 9.87 | 5 | 6332 | 0.0072 | 641.54 |
| youtube | 1134890 | 5975248 | 5.26 | 1 | 28754 | 0.0253 | 5461.30 |
| as-skitter | 1696415 | 11095297 | 13.08 | 5 | 35455 | 0.021 | 2710.63 |

Table 1: Summary of large graphs and their node degree distributions from Stanford Network Analysis Platform (SNAP) and Open Graph Benchmark (OGB). n.d. = node degree.

Figure 9 gives a straightforward comparison between APPR and power method for solving (PPR-symm) for a randomly generated sparse vector $\bar{\mathbf{y}}$. While there is considerable variability across graphs, there are several notable cases where APPR gives the better tradeoff, due to its flexible nature.

Figure 10 gives the edge ratio across a wider range of datasets. The edge ratio indicates how well a graph cluster correlates with the true labels, and depends on the specific dataset. The effect of subsampling on edge ratio is also correlated with the dataset; in effect, it indicates how much effect influencers, or other forms of edge diffusivity, are indicative of same-label or different-label behaviors.

Figure 11 shows the results for different values of $\bar{q}$, providing an ablative study.

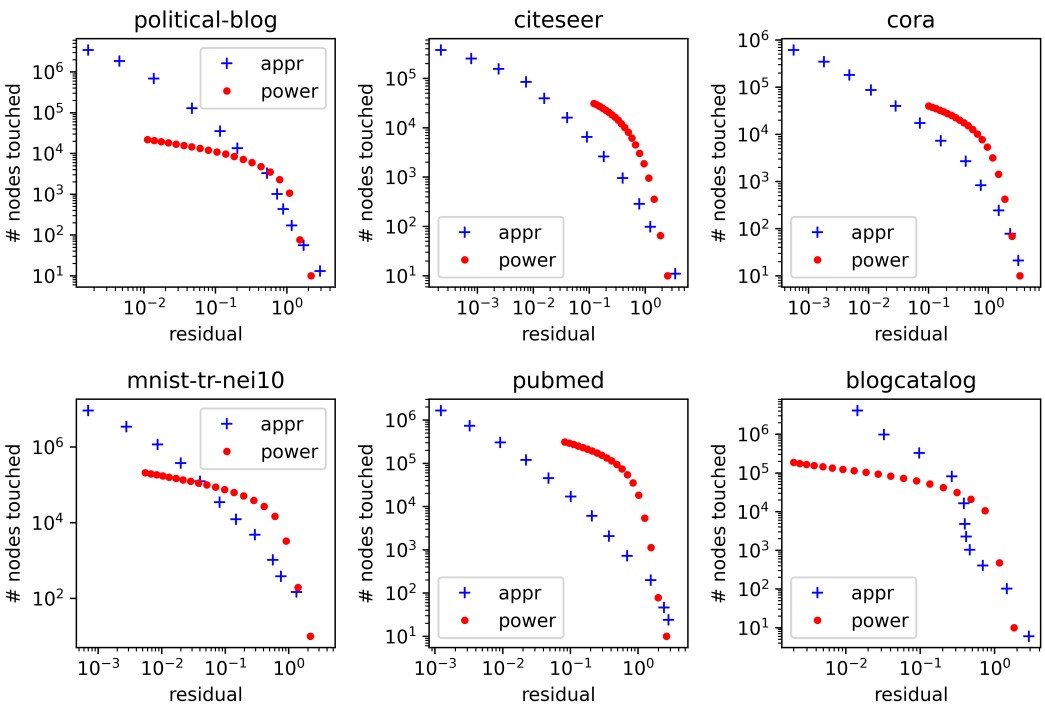

Figure 9: **Power method vs APPR.** Tradeoff curves of complexity (# nodes touched) vs performance (residual).

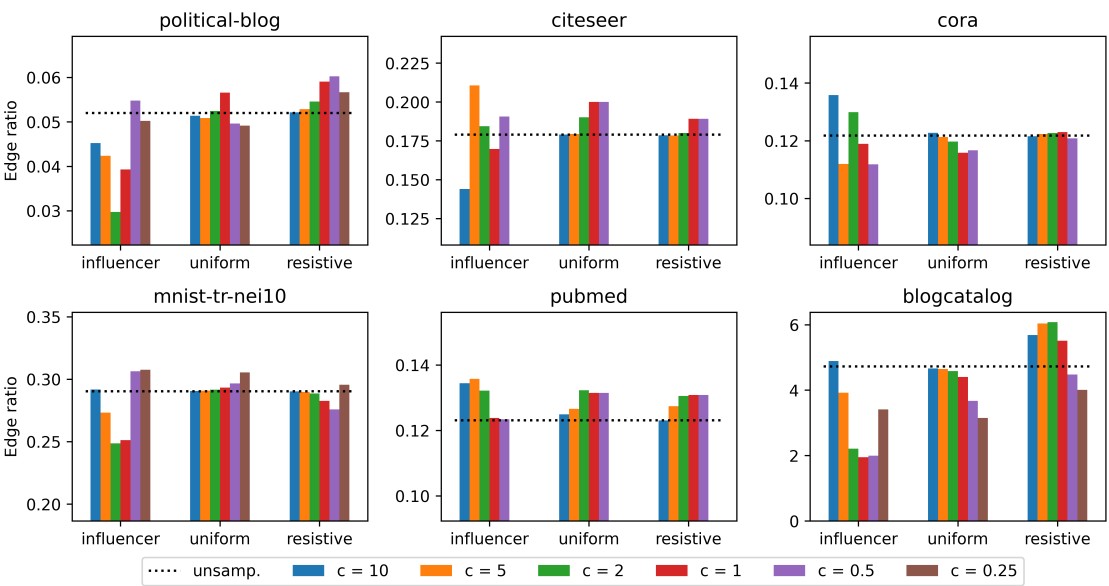

Figure 10: **Edge ratio.** This measures the proportion of edges that connect different-labeled nodes over same-labeled nodes. (Smaller is better.) Sparsifications: U = uniform, R = resistive, I = influencer. The labels show $\bar{q}$ for (I), and the corresponding sparsification rate for (U) and (R).

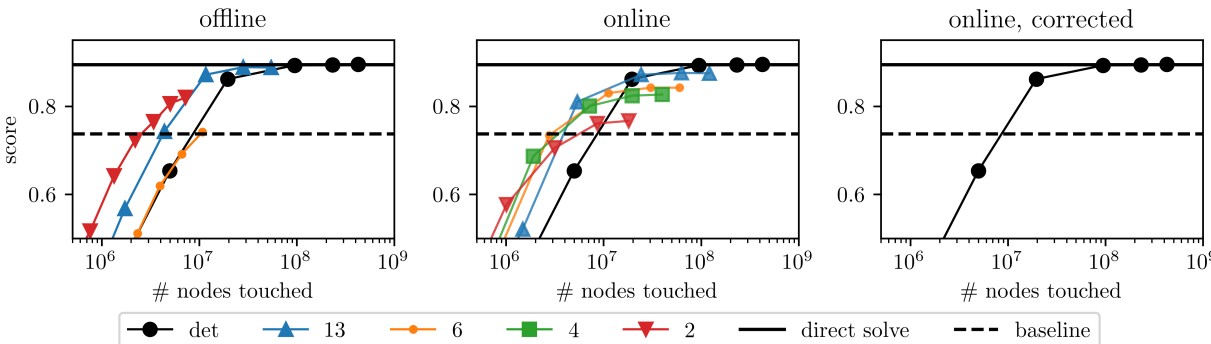

Figure 11: **Clustering performance (ablation).** det = deterministic APPR, off = offline, on = online. Key indicates $c$, and $\bar{q} = c \cdot$ median node degree. All experiments were run for the same set of $\epsilon$ values. Higher is better.

