# OpenReview forum: "Fast online node labeling with graph subsampling"
_TMLR — Rejected by TMLR_

### Review · Reviewer_ytQP · 2025-03-23

**Summary Of Contributions:**

The authors propose a graph subsampling method combined with APPR. The paper provides approximation bounds for the spectral properties of the subsampling and show a practical correlation with inverse resistive distance. The method is shown to have a powerful tradeoff between performance and complexity both theoretically and in practice.

**Audience:**

Yes

**Broader Impact Concerns:**

No concerns

**Claims And Evidence:**

Yes

**Requested Changes:**

I have no critical requested changes but the following would improve the manuscript

- further explanation for the importance of a comparison with WMA and practical scenarios where this tradeoff between a (large) complexity change and performance are important.

- further explanation for why removing the restriction on the hop radius is beneficial.

**Strengths And Weaknesses:**

Strengths

- The paper provides a strong theoretical underpinning for the proposed sampling

- I really like the added experimental ablations such as the correlation with resistive distance. These provide context to the theoretical results.

- In general, the paper is extremely comprehensive and provides evidence for the main claims. Moreover, the online sampling is a compelling proposition and will be an interesting addition to the community.


Weaknesses

- The authors point out that previous methods have used graph subsampling to improve efficiency for Graph Neural Networks. They then argue that using an APPR-inspired approach offers a more powerful tradeoff between performance and sparsity as it does not restrict the hop-radius. I think more explanation of this could be provided given this is one of the main contributions of the work. Is there an ablation that provides this evidence? It is not clear to me why the performance would be better when global (outside hop-radius) edges can be removed.

- The practical comparison to WMA is not clear as the complexity is extremely different between the two approaches. From the paper, it could be more clear in what scenarios one method would be more optimal and why this comparison is important. Given the orders of magnitude difference on the x-axis (number of nodes touched), figures 4 and 6 lose quite a bit of practical value and it is difficult to put the proposed method in perspective.

Further Questions and Points

- It is not clear to me why the edge ratio is dramatically reduced with influencer subsampling but is increased with resistive distance when they are correlated (Figure 3). Can the authors offer any explanation for this?

- Some of the figures are quite unclear. For instance, what is c in figure 3? and what dataset is being used? Additionally, as mentioned, large magnitude changes on some of the axis (Figures 4-6), makes visually interpreting differences between approaches difficult.

---

> ### Author Response · Authors · 2025-03-27
> **Response to reviewer**
>
> Thank you for your review, it has raised very interesting questions. To address some of these questions, we have added two figures to Pg 26 in the appendix of the revision.
>
> > The authors point out that previous methods have used graph subsampling to improve efficiency for Graph Neural Networks. They then argue that using an APPR-inspired approach offers a more powerful tradeoff between performance and sparsity as it does not restrict the hop-radius. I think more explanation of this could be provided given this is one of the main contributions of the work. Is there an ablation that provides this evidence? It is not clear to me why the performance would be better when global (outside hop-radius) edges can be removed.
>
> The general point we are making with regards to APPR vs Power method, and with limiting the hop radius, is that the APPR method offers more flexibility in the way it handles sparsity. In the power method, sparsity is handled through iteration truncation, e.g. we just stop multiplying $Lx$ at some point. This essentially captures the $K$-hop neighborhood of the sparsity of the initial right hand side. However, in the presence of influencers, and generally of extremely nonuniform degree distributions, the $K$-hop neighborhood could be extremely large. On the other hand, APPR, because it stops based on the residual over each node, it has the ability to stop early on some nodes and allow other nodes to continue to propagate. It still can suffer the effect of influencer nodes (and hence the need for subsampling) but at least will not carry their influencers over multiple iterations.
>
> To give an ablation, we added Figure 9. It is not a ubiquitous effect, but in many graphs and in many regimes, the APPR method is indeed balancing the node complexity and residual value.
>
>
> > The practical comparison to WMA is not clear as the complexity is extremely different between the two approaches. From the paper, it could be more clear in what scenarios one method would be more optimal and why this comparison is important. Given the orders of magnitude difference on the x-axis (number of nodes touched), figures 4 and 6 lose quite a bit of practical value and it is difficult to put the proposed method in perspective.
>
> It is true that WMA is an extremely powerful baseline, and very difficult to beat. One thing to consider is that, for many large companies, they are willing to put in the extra computation if it can improve performance, even a little. In that sense, WMA has no way of improving itself by sacrificing some computational complexity, whereas methods like APPR and even truncated power method can at least play more aggressively with this tradeoff.
>
> We agree though, that for the ONL experiments, the Relaxation method is not competitive against WMA. We leave it in because the theoretical analysis for Relaxation is more complete, where as Regularization, the more powerful method which *is* competitive against WMA, has weaker theoretical analysis.
>
>
> > It is not clear to me why the edge ratio is dramatically reduced with influencer subsampling but is increased with resistive distance when they are correlated (Figure 3). Can the authors offer any explanation for this?
>
> We revisited this example. There was a small bug in the code, which mitigated this, but even after fixing it, the effect was still there, so we ran the code for all our datasets (Fig 10). As you can see, this inverse correlation is somewhat present for political-blog and pubmed, but not there for all the plots; for citeseer, mnist, and to a lesser extent blogcatalog, the edge ratio is correlated more with resistive distance.
>
> Overall, there are some explanations for this. First, the correlation between influencer and edge inverse resistance is present, but not extremely strong, so differences in their performance over other metrics is possible. For example, edges between nodes in a fully connected subgraph will have very low resistive distance, but are not necessarily influencers.
>
> The effect of these subsampling and edge ratio also seem very data dependent. For political-blog (Fig 3), it seems that extreme influencers (more than median number of edges) should have their edges subsampled, perhaps because to become an influencer, one is somewhat indiscriminate to political party labels. However, as in the previous example, because low resistive distance can also indicate cliques, which are likely preserved in political party association, removing based on this metric is more detrimental. However, this property is less present in citeseer, where citation circles could be much smaller, and mnist, which is not based on human behavior.
>
> This is an interesting observation, and we added a paragraph of this in the discussion.

---

> > ### Author Response · Authors · 2025-03-27
> > **continued**
> >
> > **Minor issues**
> > $c$ is the factor used to determine influencer subsampling. For each node, we retain only $\bar q = c \times $ median degree of its connected edges. In Fig. 3, the dataset is political-blog, but the added figure (Fig 10) contains results for all datasets.
> >
> > We agree there are a lot of magnitude differences in Fig 4-6, but are not sure how to best mitigate this. Actually, this is quite reflective of our perceived computational differences between each experiment. Again, we stress that the main appeal of the advanced methods like APPR is in regimes where the tradeoff performance boost is most appreciated.

---

### Review · Reviewer_qnwb · 2025-04-03

**Summary Of Contributions:**

This paper considers the problem of node labeling. Based on a previous work (APPR), the authors realize that there is a benefit in terms of memory in dropping messages at random. The method relies on sampling a constant number of edges for each node to be labeled. Overall, the paper is very poorly written, with no clear structure. The story line of the paper lacks any causal structure -- there is no problem formulation, very little explanation of what the main contributions of this work is. The algorithms are dropped out of nowhere (and arranged very poorly on the page), and there is no structure of which algorithm is new, and which one comes from another paper. On top of that, comments are made throughout the paper with no proper citations.
I believe this paper, in its current form should be rejected, it is almost unreadable.

**Audience:**

No

**Broader Impact Concerns:**

N/A.

**Claims And Evidence:**

No

**Requested Changes:**

See above.

**Strengths And Weaknesses:**

# Strengths

[S1] The paper is considering an important problem. There exists interest in the community for efficient methods to for the node labeling problem.

# Weaknesses

[W1] The paper assumes that the reader is a complete expert on the field. The paper does not provide any explanation on the problem being solved, the existing solutions and the link between them. The authors should try to align the way they write the paper to any other paper in this journal. I have been reviewing papers for TMLR, and honestly, the structure of this paper is the most confusing and difficult to follow of all the ones I have read. The authors should follow the structure of TMLR paper by using sections as: Problem formulation, Algorithmic explanations, and Theoretical parts properly. In other words: (1) define the problem you want to solve, (2) explain how, (3) show your algorithm, (4) introduce the theory.

In this paper, there are comments about other methods, other works algorithms, and other people's theory all over the paper. It is imposible to follow. For example:

> "In this paper, we extend the work of (Andersen et al., 2006; Zhou et al., 2023) to graphs with unfavorable node degrees, using edge subsampling. We propose a simple approach: for a threshold q¯, we identify all nodes with degree exceeding q¯, and subsample their neighboring edges until they have ≤ q¯ neighbors. The remaining edges are reweighted such that the expected edge weights are held consistent. In offline graphs, sparsifications of this kind is O(n) where n is the number of nodes; however, the dependency on n is removed when done online."

No citations for  "In offline graphs, sparsifications of this kind is O(n) where n is the number of nodes"
No proper explanation" however, the dependency on n is removed when done online."

[W2] There are plenty of grammar mistakes. The writing quality of the paper is very low, and this makes the paper imposible to follow:
> "However, the disadvantage of this strategy is that the stochasticity leads a high variance between each trial; thus, we implement present a mechanism for grounding the residual at each iteration, to reduce this variance"

[W3] The explanation and mathematical rigor of this paper is very low.
- Equation 1: take the lim, or explain what your notation is
- "nation steps in the Push substep" What is this? The push step was never introduce. Imposible to follow.
- In Alg 1, nothing is explained. How is the reader supposed to know what is going on.
- Lemma 2.1 cannot be understood.
- Figure 1, the top 1 are dots, and the bottom one are lines. Why?
- Citation needed “In electrical engineering, this measures the effective resistance between two junctions i and j in a network of resistors, whose weights form the weights of the graph (Ai,j = 1/ri,j ).” Also, why is this relevant?
- Theorem 3.1 cannot be understood.


[W4] The numerical experiments are too simple and do not showcase a lot. There is only one baseline considered and for very small graphs. This graphs can be run in a single GPU.

---

> ### Author Response · Authors · 2025-04-05
> **response to reviewer**
>
> > there is no problem formulation, very little explanation of what the main contributions of this work is.
>
> Thank you for your feedback. This paper builds on the APPR method, which has been widely used for solving linear systems and graph-based inference tasks. While APPR has previously been applied to online node labeling, our work proposes a stochastic extension that subsamples high-degree nodes. Our contributions include:
>
> 1. A variance-reduced stochastic variant of APPR tailored to skewed-degree graphs,
>
> 2. Theoretical analysis on approximation and convergence, and
>
> 3. Empirical results across three settings: solving linear systems, online node labeling, and graph clustering.
>
> This is described in our introductory paragraph under "contributions"
>
> > The authors should follow the structure of TMLR paper by using sections as: Problem formulation, Algorithmic explanations, and Theoretical parts properly...
>
> We appreciate the suggestion. We recognize the value of a standard structure. Our intent was to organize the paper around a methodology-driven narrative—starting from motivating APPR, describing our extension, and then analyzing it both theoretically and empirically. However, we agree that clarifying which parts are new and aligning more with TMLR structure (e.g., explicitly delineating problem formulation, algorithm, and theory) will improve readability.
>
> > No citations for "In offline graphs, sparsifications of this kind is O(n)...”
>
>
> Thank you for pointing this out. Our statement was intended as a high-level intuition rather than a formal claim. Offline sparsification typically requires accessing all \(n\) nodes, while online variants operate only on a queried subset. We've revised the sentence to clarify this distinction and removed ambiguity regarding theoretical claims.
>
> > [W2] There are plenty of grammar mistakes...
>
> Thank you. We’ve carefully proofread the manuscript and corrected numerous grammar issues. We’ve also used automated tools to catch any remaining typos and will continue refining clarity in future revisions.
>
> > Equation 1: take the lim, or explain what your notation is
>
> We intended Equation (1) to reference the von Neumann series expansion, which is standard in linear algebra when expressing a matrix inverse via a geometric series. We have added a short explanation to clarify this interpretation and its connection to the solution of PPR systems.
>
> > "nation steps in the Push substep"... push step was never introduced
>
> Thank you for catching this. We've now added an explicit sentence prior to Section 2.1:
> *“The steps within the while loop are collectively referred to as the Push operation.”*
> This aligns with terminology in Andersen et al. (2006), though we agree it should have been clearly introduced.
>
> > In Alg 1, nothing is explained...
>
> Since Algorithm 1 is taken from prior work, we opted to summarize it briefly to save space, especially given page constraints. That said, we've added clarifying language to explain that Alg. 1 solves a linear system via a localized variant of the power method, and highlighted how it differs from standard iterative solvers via its residual-based truncation strategy.
>
> > Citation needed for effective resistance... why is this relevant?
>
> This is a well-established concept in graph analysis and is discussed extensively in Spielman and Srivastava (2008), which we have cited. The relevance of this idea is also made clear in their work, where they demonstrate that sparsification based on scaling by effective resistance leads to optimal spectral bounds. Our aim in including this experiment is to draw a connection between our sparsification strategy—downsampling high-degree nodes—and the physical intuition that edges connected to high-degree nodes often exhibit low resistance.
>
> > [W4] The numerical experiments are too simple...
>
> We appreciate this point and agree that our experiments are limited in scale. Our current computational resources constrained us from running larger-scale tests. However, we note that larger graphs tend to be sparser, which generally improves the scalability of our method. Moreover, our goal is to demonstrate the conceptual value of our method and its consistency, even on modestly sized graphs.
>
> Also, we clarify that our experiments were run entirely on CPUs, which is common in graph-based applications, particularly where memory locality and sparsity are the focus rather than raw throughput.

---

> ### Author Response · Authors · 2025-04-05
> **Response continued**
>
> >Lemma 2.1 is confusing
>
> We appreciate the comment. Lemma 2.1 summarizes a core result from the original APPR work, showing that the sparsity of the iterates remains bounded and localized—a key advantage of these methods. Specifically, the lemma highlights that the support of the iterates expands monotonically and remains within the support of the final solution:
>
> $\mathbf{supp}(x^{(t)}) \subseteq \mathbf{supp}(x^{(t+1)}) \subseteq \mathbf{supp}(x^*).$
>
> While this is not our own contribution, we included it to help motivate the relevance of APPR and why it lends itself naturally to efficient implementation. To avoid redundancy, we defer detailed proofs to Andersen et al. (2006), where it is presented and discussed in depth.
>
>
>
> > Theorem 3.1 is confusing
>
> Thank you for pointing this out. Theorem 3.1 applies a standard subgaussian concentration bound to quantify the behavior of our sampling scheme. It follows the same general structure as prior work in graph sparsification—particularly Karger (1994), Benczúr & Karger (2015), and Spielman & Srivastava (2008). While the setting differs slightly due to our use of degree-based sampling, the overall approach mirrors those well-established results, which are widely used in the literature.

---

### Review · Reviewer_apPM · 2025-04-17

**Summary Of Contributions:**

The paper studies graph sparisfication for the purposes of node classification. While a good portion of their contributions is in experiments, I would like to remark on the theoretical aspects where I got immediately stuck. I stopped reading at Theorem 3.1 since this theorem already was very strange to me. If the authors can address my concerns I would be happy to provide feedback on the rest of the paper.

Theorem 3.1 is their main contribution of 'offline sparsification'. It is used as a key building block for their later contribution of online sparsification. Theorem 3.1 says the following: first we pick a threshold $\overline{q}$. All nodes with degree exceeding $\overline{q}$ have their edges sub sampled and re-weighted until their degree is $\le \overline{q}$.

There are several major issues here which is why I stopped reading:

1) What is the probability of sub sampling the edges? Is it uniform?
2) The probability statement seems meaningless. Note that $d_{\max}^2$ is in the denominator of the fraction inside the $\exp$ function. In particular their concentration bound is of the form $\exp(- \overline{q}^2/d_{\max}^2)$. This statement only provides non trivial guarantees when $\overline{q} \gg d_{\max}$ which means the threshold is doing nothing. This is because otherwise the term inside the exponential is smaller than $1$ so the statement is saying something like 'The probability of deviation is bounded by 1' which is meaningless.
3) There is a reason why prior work of Karger et al and Srivastava et al is so famous. They are able to union bound over *exponentially many* vectors x (in the case of Karger's cut sparsification) or over all x in the case of spectral sparsification of Srivastava et al. Theorem 3.1 of the submission only handles a single $x$ which is trivial. The whole point of sparsification is to preserve `all possible quantities' (either all cut values or all Laplacian quadratic forms). Thus even ignoring the fact that the probability statement for even one x is already not saying much, Theorem 3.1 cannot union bound over a large class of vectors x. Note that prior works can be done in linear time as well as in a single pass streaming setting.
3) There seems to be a major misunderstanding of the theorem statement by the authors themselves. They say their theorem is most meaningful when $d_{\max} \gg \overline{q}$ which I pointed out above would lead ot something trivial. there is also a reason why all prior works with meaningful theoretical guarantees have a $1/\epsilon^2$ dependency. This is unavoidable in sampling in general! It is known that for both node and spectral sparsification, $\Omega(n/\epsilon^2)$ edges in total *need* to be sampled (https://arxiv.org/abs/1712.10261). So I find that the fact that the total sample complexity being $O(n/\epsilon)$ a major flaw in this work since it implies there is nothing interesting theoretically happening here.

Another note: The figures are of really low quality. I would recommend saving them with higher dpi or saving them as pdfs and then adding them to the latex document.

**Audience:**

Yes

**Claims And Evidence:**

No

**Requested Changes:**

Can the authors explain the points raised above?

**Strengths And Weaknesses:**

See above. I am already stuck at a major weakness in the early parts of the paper.

---

> ### Author Response · Authors · 2025-04-19
>
> > Theorem 3.1 is their main contribution of 'offline sparsification'. It is used as a key building block for their later contribution of online sparsification.
>
> Actually we may say that the theorems are all a bit orthogonal. Theorem 3.1 is the main building block for influencer sparsification as a concept, but the convergence bounds can be applied to any sparsification scheme.
>
>
> > Theorem 3.1 says the following: first we pick a threshold. All nodes with degree exceeding  have their edges sub sampled and re-weighted until their degree is . What is the probability of sub sampling the edges? Is it uniform?
>
> In our analysis, it is uniform. In practice, we also experiment with weighted by edge weight, or by resistive distance value.
>
> > The probability statement seems meaningless. Note that  is in the denominator of the fraction inside the  function. In particular their concentration bound is of the form . This statement only provides non trivial guarantees when  which means the threshold is doing nothing. This is because otherwise the term inside the exponential is smaller than  so the statement is saying something like 'The probability of deviation is bounded by 1' which is meaningless.
>
> We are very sorry about this, we went back and investigated this issue and realized it was a bug in our proof. If you look at Lemma B4 and the following corollary, the term $S(\frac{\bar q}{\max\{d_i,d_j\}})$ disappeared after a few steps, but was critical to, exactly as you point out, make sure this bound is not vacuous. This seemed to have also caused  some errors in our subsequent analysis. However, we stress that since this is orthogonal to the optimization component, no other result is affected by this.)
>
> We acknowledge that we did not look at the implications of this bound closely enough. We were focused on computing the closest subgaussian parameter possible, without realizing that because subgaussian is so much weaker than variance, that indeed, you are right, in the low $p_{i,j} = \bar q/\max\{d_i,d_j\}$ regime, our bound is also not competitive, although in practice in that region things are great. But the bound is no longer vacuous, as in the larger $p_{i,j}$ regime, looking at Fig 8, indeed our bound (the green line) is below $p^2$, so in that area our bound is nonvacuous.
>
> However, given that we still operate in the low $p_{i,j}$ regime, we reinvestigated everything and realized that using the subgaussian parameter indeed did not provide the most competitive bound. We also realized that the normalized Laplacian matrix is bounded in values, so we replaced it with the variance parameter. The resulting bound is what is now included in the current version, which is nonvacuous when $|S_I| << n$ (e.g. the number of nodes whose degree exceeds $\bar q$ is very small, e.g. a few influencers.)
>
>
> > There is a reason why prior work of Karger et al and Srivastava et al is so famous. They are able to union bound over exponentially many vectors x (in the case of Karger's cut sparsification) or over all x in the case of spectral sparsification of Srivastava et al. Theorem 3.1 of the submission only handles a single
>  which is trivial. The whole point of sparsification is to preserve `all possible quantities' (either all cut values or all Laplacian quadratic forms).
> Thus even ignoring the fact that the probability statement for even one x is already not saying much, Theorem 3.1 cannot union bound over a large class of vectors x. Note that prior works can be done in linear time as well as in a single pass streaming setting.
>
> Our goal is to produce the same flavor of bound as  Srivastava et al. Note that if you normalize $\epsilon$, there is not much dependence on $\|x\|$ in the bound (this is in our updated version). Although we agree our statements regarding our bound's superiority was premature, and we have removed them.
>
>
>  >It is known that for both node and spectral sparsification,
>  edges in total need to be sampled (https://arxiv.org/abs/1712.10261). So I find that the fact that the total sample complexity being
>  a major flaw in this work since it implies there is nothing interesting theoretically happening here.
>
> We should emphasize that it is not our intention to beat the sample complexity of spectral sparsification, but to provide a computationally cheap alternative with a comparable sample complexity. We believe this is still present.
>
> >Another note: The figures are of really low quality. I would recommend saving them with higher dpi or saving them as pdfs and then adding them to the latex document.
>
> Thanks, we have updated the new pdf with higher dpi.

---

> > ### Comment · Reviewer_apPM · 2025-05-06
> > **Folow up on Theorem 3.1**
> >
> > I am still not convinced Theorem 3.1 is saying anything. First, the statement has a typo: the $\epsilon$ is being divided by a term but that term is missing. Let's ignore this for now.
> >
> > The lemma is saying the probability that $x^TLx$ deviates from $x^T\tilde{L}x$ by an $\epsilon$ term, where $\tilde{L}$ is the laplacian of the sampled graph, is bounded by $\exp(-\epsilon^2/|S_I|)$ roughly (there is another term in the denominator but let's ignore this for sake of simplicity), where $|S_I|$ is the number of edges adjacent to `influential' nodes (one of the endpoints has degree $\ge \overline{q})$.
> >
> > Note that their $L$ is the normalized laplacian so all of its eigenvalues are bounded by $O(1)$ in magnitude. Let $x$ be a unit vector by scaling. Then $|x^TLx|<= O(1)$ always. So the meaningful range of $\epsilon$ is also $O(1)$. But then the statement is entirely meaningless for any intersting range of $|S_I|$. We expect this to be super constant (can be as large as $n$). Even if this is say $\log n$ (which is not realistic ...), the probability bound won't give you anything meaningful! E.g. take $\epsilon = 1$ even and $|S_I| = 1000$. Then the probability is $\exp(-1/1000) \approx 1$. This gets even closer to one as $|S_I| = \omega(1)$....
> >
> > Theorem 3.1 should be changed or entirely removed if the authors want to convey anything meaningful.

---

> ### Author Response · Authors · 2025-05-06
> **subsampling regime**
>
> Thanks for looking over the theorem so closely. We agree with the statements you made, but it is important to note that indeed, the intent of this bound is for $|S_I|\ll n$, even moreso than $|S_I|\ll\log(n)$. This is under the observation that for very large datasets, as the graph grows, the degree distribution becomes more and more skewed, so that even thresholding at, say, $\bar q = $ 5*median degree provides significant memory saving, especially if the graph is distributed over multiple machines. This is supported by Fig. 1, which indeed shows that extreme degree distributions are common in practice. We were not able to reach this value of $\bar q$ in our experiments, as they are still too small-scale, but we believe that this theoretical bound demonstrates something useful in the larger data regime.
>
> We also fixed the typo, thanks.

---

> > ### Comment · Reviewer_apPM · 2025-05-07
> > **Follow up to the authors**
> >
> > There maybe a misunderstanding. The larger $|S_I|$ is, the more meaningless the theorem statement is. My point was that as long as $|S_I|$ is super constant (even $\log(n)$ for example), the theorem gives a probability bound very close to $1$, which is a trivial statement. My understanding is that the theoretical statement does not demonstrate anything useful. To summarize, we can normalize $\|x\|_2 = 1$. Then $x^TLx = O(1)$ always since $L$ is the normalized laplacian matrix. So the meaningful range of $\epsilon$ is also constant, but the denominator in the probability statement is super large (much larger than a constant), so the right hand side of the probability bound is going to 1.

---

> ### Author Response · Authors · 2025-05-07
>
> So $S_I$ are the set of nodes whose edges have been subsampled. In an extreme power distribution scenario, $|S_I|\ll n$. We are considering the regime of very small $|S_I|$, not very big ones.
>
> By the way, the way we had corrected that typo wasn't quite right; we were trying to normalize by $\|x\|_\infty$, but it seems to lead to some confusion, so we changed it back to the unnormalized version
>
> $$ Pr( x^T {\tilde {L_I}} x  - x^T L x \geq \epsilon )  \leq \exp( -\frac{\epsilon^2}{\|x\|_\infty^4 |S_I| + (2/3)} ).$$
>
> Suppose $x$ is normalized, e.g. $\|x\|=O(1)$. Then as $|\mathcal S_I|\to 0$, the right hand side converges to $\exp(-3\epsilon^2/2)$ which we believe is meaningful, in the regime, as you rightly pointed out, of $0<\epsilon<2$.

---

> ### Comment · Reviewer_apPM · 2025-05-07
> **Theorem 3.1 still lacks any substance**
>
> I still claim that Theorem 3.1 is still not meaningful / lacks any substance.
>
> First, $S_I$ is defied by the authors (see bottom of page 19) to be the set of edges where at least one end point has degree larger than the threshold. In particular, $S_I$ contains all the edges of the nodes with degree larger than the threshold. Note that this value is an integer so it cannot "go to 0" as the authors state in the previous reply. In addition, it must be at least the max node degree which even in the 'real world' graphs is much larger than $\log(n)$ (and should be going to infinity with the size of the graph in any interesting graph model). Thus, my point above still stands: the theorem is saying the probability of deviation is bounded by $\exp(-\epsilon^2 / |S_I|)$ where $\epsilon = O(1)$ is the only meaningful regime, but $|S_I|$ is very large (and in the natural cases is going to infinity!). Thus this is a totally meaningless probability guarantee since this quantity is approaching $1$. I do not want to keep repeating this point but having a large quantity in the denominator of  a fraction $a/b$ in $\exp(-a/b)$ is `bad'.
>
> Second, which is something that I pointed out initially which the authors ignored, is that there is a reason why prior work of Karger et al and Srivastava et al is so famous. They are able to union bound over exponentially many vectors x (in the case of Karger's cut sparsification) or over all x in the case of spectral sparsification of Srivastava et al. Theorem 3.1 of the submission only handles a single which is trivial. The whole point of sparsification is to preserve `all possible quantities' (either all cut values or all Laplacian quadratic forms). Thus even ignoring the fact that the probability statement for even one x is already not saying much, Theorem 3.1 cannot union bound over a large class of vectors x. Note that these prior works can be done in linear time as well as in a single pass streaming setting.
>
> Overall having some math for the sake of having math is not a good practice.

---

> ### Author Response · Authors · 2025-05-07
>
> You are right, it must be at least 1, but it is **not** related to the max degree itself. As previously stated, $S_I$ are the set of nodes whose degree exceeds our threshold. For very high degree graphs, if that threshold is a multiple of the median node degree, this set is **very small**.  There is no reason why this quantity must be related to $O(n)$ or even $O(\log(n))$. This is a **key misunderstanding**.
>
> As regards to the prior results, the quadratic form we have given is **equivalent** to that given in the Srivastava et al paper. Since $x$ can be eigenvectors, this means that spectral properties are implicitly preserved -- that is exactly what they are saying as well. You are correct that the bound is somewhat loose unless $n$ is very large and $|S_I|$ is very small, but this does not negate the correctness of our statement, or the value of the statement in this specific graph regime.
>
> It is not correct to say this is math for the sake of math. It is providing a theoretical bound that 1. is appropriate for the method we are giving and 2. has value in the asymptotic regime. This is very much in line with these kinds of results in many existing papers.

---

> > ### Comment · Reviewer_apPM · 2025-05-07
> > **Follow up to authors**
> >
> > What is the definition of $S_I$ that you are using? On page 19 in your pdf it is stated as the set of edges where at least one endpoint has degree larger than the threshold. The threshold you set must be smaller than the max degree (do you see why?) so by your own definition, $S_I$ will at least contain the edges of the vertex realizing the max degree and hence $|S_I|$ will be at least as large as the max degree. In your experiments, this is a very large quantity and in any interesting random graph model, this will go to infinity as $n$ does.
> >
> > The quadratic form that you give is also not equivalent or even comparable in any meaningful way to the prior results at all. Again as I mentioned above, prior results eg those by Srivastava et al are able to *union bound for every $x$ simultaneously*. In simper terms, there is a 'for all x' inside the probability statement itself. The result in the submission at best can only give a bound for a single vector $x$.

---

> > > ### Author Response · Authors · 2025-05-07
> > > **Apologies**
> > >
> > > Sorry, you are right. It has been a while since we started working on this paper, my apologies. Give us a day or so to review this issue and get back to you. I still believe there should be a concentration inequality applicable here, in the same spirit as Srivastava et al., but you raise valuable and valid concerns.

---

> > > > ### Author Response · Authors · 2025-05-07
> > > > **Revision uploaded**
> > > >
> > > > Thanks for the discussion. We have revisited our bound to more carefully understand the interplay between $\epsilon$ and $S_I$. We made the following changes.
> > > >
> > > > 1. We added a Chernoff inequality approach, which can be used to eliminate dependency on $|S_I|$ altogether. The resulting bound indeed depends on the structure of $x$, but under randomly generated Gaussian i.i.d. $x$, leads to a nontrivial bound for large $n$. This should settle questions as to the meaningfulness of the right-hand-side values. (This is term A)
> > > >
> > > > 2. We also inserted $|S_I|$ in by approaching the Chernoff inequality bound in two different ways. (This is term B)
> > > >
> > > >
> > > > $$Pr\left(\left|x^T \tilde{L_I} x - x^T L_I x\right| \geq \epsilon\right)
> > > > \leq 2 \min( A,B)
> > > > )$$
> > > > where
> > > > <div class="math">
> > > > \[
> > > > A = \exp(-\frac{ \epsilon^2 }{8 \|x\|_\infty^2 \|x\|_2^2}) , \qquad
> > > > B = \exp(-\frac{\epsilon^2}{8\|x\|_\infty^4|{S_I}|}).
> > > > \]
> > > >  </div>
> > > >
> > > >
> > > >
> > > >
> > > > Specifically, if $x\in \mathbb R^n$ is i.i.d. , then the ratio
> > > >
> > > > <div class="math">
> > > > \[
> > > > \mathbb E[\frac{\|x\|_2}{\|x\|_\infty}]
> > > > O(\sqrt{n/\log(n)})
> > > > \]
> > > > </div>
> > > >
> > > > This leads to the following reduction, assuming $\|x\|_2=O(1)$ (or more generally, that $\epsilon / \|x\|_2 = O(1)$):
> > > >
> > > > $$Pr\left(\left|x^T \tilde{L_I} x - x^T L_I x\right| \geq \epsilon\right) =
> > > > O(\min\{e^{-\frac{ \epsilon^2 n}{8\log(n)  }},e^{-\frac{ \epsilon^2 n^2}{8 \log(n)^2|\mathcal S_I|}}\}).
> > > > $$
> > > >
> > > > The first term indeed is meaningful when $n$ is large, and is the main revision. However, the second term also pops out when using the Bernstein bound, and when using this new Chernoff approach, if we insist on the result depending on $|S_I|$ (which we do, in order to illustrate the effect of subsampling). Specifically, we now see that   the second term  becomes active if $|\mathcal S_I| < O(n/\log(n))$. This is a bit smaller than we wanted (extreme influencers being closer to $O(n)$, but we feel it is still a valuable regime.
> > > >
> > > > Clearly a weakness (and a new observation) is that the meaningfulness of the result depends a lot on the structure of $x$ itself. However, assuming its uniformity is at least that of a Gaussian generated vector is not a very strong assumption, so we hope it will alleviate some concerns.

---

> > > > > ### Comment · Reviewer_apPM · 2025-05-07
> > > > > **Follow up on authors**
> > > > >
> > > > > I still claim that Theorem 3.1 is not saying any thing meaningful.
> > > > >
> > > > > - The new term the authors added of $\exp(- \Theta(\epsilon^2))$ (say in the case that $\|x\|_2 = 1$ which is without loss of generality by normalizing) is totally useless. This term does not go to zero as the sampling gets better! This means that the sampling algorithm is not useful. The probability bound is just a fixed value always. Also note that for $\|x\|_2 = 1$, their term $2\exp(-\epsilon^2/8)$ is only less than $1$ for $\epsilon > 2$ which is not useful because $x^TLx$ is always bounded by $2$ since $L$ is a normalized laplacian (psd matrix with max eigenvalue $2$).
> > > > >
> > > > > - Assuming the vector $x$ is Gaussian is not interesting. Historically, an important subset of vectors $x$ has entries $\pm 1$ since this corresponds to cuts in the graph. (Note that historically first we had cut sparsification of graphs due to Karger et al and then the stronger spectral sparsification e.g. Srivistava et al which captures cuts as a special case). Also if you assume $x$ is Gaussian, likely one can fix any matrix $A$ and get strong concentration of $x^TAx$ just from the Gaussian assumption on $x$ itself (e.g. concentration of gaussian quadratic forms).
> > > > >
> > > > > - Third, as I have pointed out, sparsification is only interesting when one can union bound over a large collection of vectors $x$. Historically, the cut sparsification results were able to provide a guarantee for all $\pm 1$ vectors simultaneously and later the spectral sparsification results were able to union bound over all vectors $x$. This result cannot extend past 1 vector.

---

> > > > > > ### Author Response · Authors · 2025-05-07
> > > > > >
> > > > > > The important new observation is that the ratio of
> > > > > >
> > > > > > <div class="math">
> > > > > > \[\rho = \|x\|_2/\|x\|_\infty\]
> > > > > > </div>
> > > > > >
> > > > > >  depends on the uniformity of the values in $x$. Note that if $x_i = \pm 1$, then there is *maximum* uniformity, and $\rho = \sqrt{n}$, and actually our bound is *stronger* -- the log terms drop out and it decays as $O(\exp(-n\epsilon))$. Note this is holding for *all cut vectors*. So, the Gaussian assumption is *weaker* than assuming all cut vectors. In fact, the only time that $\rho = 1$ is if $x$ is 1-sparse, which is not an interesting regime for us.
> > > > > >
> > > > > > I am not sure why you say it cannot extend past 1 vector; perhaps there is a misunderstanding here. Can you please clarify?

---

> ### Comment · Reviewer_apPM · 2025-05-08
> **Follow up**
>
> Note there is no ratio in the statement of Theorem 3.1 in the right hand side. There is a product $\|x\|_{\infty}^2 \|x\|_2^2$ together in the denominator. For unit vectors $x$, this value is $1$ and the new term is useless (it gives something larger than $1$ for all $\epsilon \in (0, 2)$.
>
> For vectors with larger $\|x\|_2$, the denominator grows and the probability bound approaches $1$ rapidly e.g. for $\|x\|_2^2 = n$ which is for cut vectors and gaussian vectors you get something on the order of $\exp(-1/n) \approx 1$.
>
> Prior work have statements of the form $P(\forall x ....)$. This is what I mean by their results holds for all vectors $x$ simultaneously. This was a point I made in the very first review.
>
> I think I have made my final evaluation. I thank the authors for the discussion.

---

> > ### Author Response · Authors · 2025-05-08
> >
> > The reason we pick |x|_2=1 is because the left hand side is normalized when |x|_2 = 1. Otherwise, we can repeat the entire argument with $\tilde \epsilon = |x|^2_2 \epsilon$.
> >
> > But ok, let's consider the set of cut vectors. Then $\|x\|_2 = \sqrt{n}$
> >
> > and $\|x\|_\infty = 1$. Then
> >
> > <div class="math">
> > \[
> > Pr\left(\left|x^T \tilde{L_I} x - x^T L_I x\right| \geq  \epsilon\right)
> > \leq 2  \exp(-\frac{ \epsilon^2 }{8 n})  \iff Pr\left(\left|x^T \tilde{L_I} x - x^T L_I x\right| \geq  n\epsilon\right)
> > \leq 2  \exp(-\frac{n \epsilon^2 }{8 })
> > \]
> >  </div>
> >
> > So we can clearly weaken our statement to say for all cut vectors (e.g all vectors where $x_i=\pm 1$), the bound is $O(\exp(-n\epsilon^2))$.

---

> > > ### Comment · Reviewer_apPM · 2025-05-08
> > > **Follow up**
> > >
> > > No, your result does not allow you to union bound over all cut vectors. Because there are $\binom{n}{n/2} =  \Theta(2^n/\sqrt{n})$ possible cuts. To union bound, you would need to have $2\exp(-n\epsilon^2/8) \cdot 2^n < 1$ which requires $\exp(-n\epsilon^2/8 + n \log(2)) < 1/2 $ which does not hold even for $\epsilon = 1$. But $\epsilon = 1$ is not interesting since you are not even preserving any degrees. But of course this has to be the case because it is known that we need to sample $\Omega(n/\epsilon^2)$ edges for cut sparsification! This is a well known result (https://arxiv.org/pdf/1712.10261).
> > >
> > > Also note that fundamentally your bound cannot make sense because it does not depend at all on the threshold that you set (your new bound has no dependence on $\overline{q}$).

---

> > > > ### Author Response · Authors · 2025-05-10
> > > >
> > > > Ok, we have looked into your statements more carefully. We understand now what you mean: while our bound showed concentration *pointwise* for $x$, it did not show concentration *uniformly*. Therefore, we cannot say that a spectral bound holds.
> > > >
> > > > We have revised this under the following approach. First, we adjust some of the analysis for $x$ a Gaussian random vector, normalized distribution. We then use a JL-like result from Vershynin to show that if such a bound is present for a Gaussian random vector, then with high probability it is true spectrally. This is not exactly uniform convergence, but it is a spectral bound.
> > > >
> > > > We believe this is in the same vein now as the result offered by Srivastava et al, but we continually welcome valuable feedback on the matter, as it can strengthen the paper.
> > > >
> > > > We would also like to stress that 1. our pointwise result was not *wrong*, simply less strong than Sriviastava et al (though now we have strengthened it) and 2. is not the *main* contribution of the paper, but merely one of several contributions. We hope this will be taken into account as assessments are being made.

---

### Author Response · Authors · 2025-04-19
**Changes in paper**

Thank you to all the reviewers for their thoughtful reviews. They have helped us find important areas of improvement, and we have made the following changes in the revision

 - A major update in the proof of Theorem 3.1, which provides the same overall exponential sample complexity, but with constants that are more meaningful to our regime, and an adjustment to the conclusions.

 - Some extra examples of the resistive distance experiments, and a comparison between the general APPR and power method for a handful of problems, for a more comprehensive view.

 - Higher quality figures

---

### Decision · Action_Editor_6GAw · 2025-06-05

**Recommendation:** Reject

**Comment:**

As discussed above, all three reviewers recommended "reject" or "leaning reject". There are concerns about the correctness and meaningfulness of an important theoretical result. Concerns have been expressed about the clarity of the paper, with a request that there be clearer identification of the novel contribution. The experiments provided evidence of marginal improvement in a limited setting, raising questions about the significance of the results and whether they generalize to other tasks and algorithms.

**Audience:**

Some individuals would be interested in knowing the findings of the paper, although it would be more likely to interest many more individuals if efforts were made to improve the clarity of the paper. The issue of the correctness and meaningfulness of the theoretical results also needs to be addressed before the paper is distributed to a broader audience.

**Claims And Evidence:**

The reviewers all expressed significant concerns about the paper. These included: (i) the paper makes a relatively minor contribution, does not clearly express the novelty, and is difficult to read and follow; (ii) there is an absence of mathematical rigor, leading to some claims not being substantiated; (iii) the experiments are limited and demonstrate only a marginal improvement in a small study. One of the reviewers raised a significant concern about one of the theoretical results, and although the authors attempted to address this during the response period, making multiple revisions of the paper, the changes are sufficiently substantial that the new version of the paper really needs another review process to allow the reviewers adequate time to assess the correctness of the revised result.

WIth these concerns in mind, and based on my own reading of the paper, it is not clear that the claims are adequately supported by convincing and clear evidence.

**Resubmission Of Major Revision:**

The authors may consider submitting a major revision at a later time.